

# The biogeophysical effects of idealized land cover and land management changes in Earth System Models

Steven J. De Hertog[1], Felix Havermann[2], Inne Vanderkelen[1], Suqi Guo[2], Fei Luo[3,4], Iris Manola[3], Dim Coumou[3,4], Edouard L. Davin[5,6,7], Gregory Duveiller[8], Quentin Lejeune[9], Julia Pongratz[2,10], Carl-Friedrich Schleussner[9], Sonia I. Seneviratne[11], and Wim Thiery[1]

[1]Vrije Universiteit Brussel, Department of Hydrology and Hydraulic Engineering, Brussels, Belgium
[2]Ludwig-Maximilians-University Munich, Department of Geography, Munich, Germany
[3]Vrije Universiteit Amsterdam, Institute for Environmental studies, Amsterdam, Netherlands
[4]Royal Netherlands Meteorological Institute (KNMI), De Bilt, Netherlands.
[5]Wyss Academy for Nature, University of Bern, Bern, Switzerland.
[6]Climate and Environmental Physics division, University of Bern, Bern, Switzerland.
[7]Oeschger Centre for Climate Change Research, University of Bern, Bern, Switzerland
[8]Max-Planck-Institute for Biogeochemistry, Jena, Germany
[9]Climate Analytics, Berlin, Germany
[10]Max Planck Institute for Meteorology, Hamburg, Germany
[11]ETH Zurich, Institute for Atmospheric and Climate Science, Zurich, Switzerland.

**Correspondence:** Steven De Hertog (steven.de.hertog@vub.be)

**Abstract.** Land cover and land management change (LCLMC) has been highlighted for its critical role in mitigation scenarios, both in terms of global mitigation and local adaptation. Yet, the climate effect of individual LCLMC options, their dependence on the background climate and the local vs. non-local responses are still poorly understood across different Earth System Models (ESMs). Here we simulate the climatic effects of LCLMC using three state-of-the-art ESMs, including the Community

Earth System Model (CESM), the Max Planck Institute for Meteorology Earth System Model (MPI-ESM) and the European Consortium Earth System Model (EC-EARTH). We assess the LCLMC effects using four idealized experiments: (i) a fully afforested world, (ii) a world fully covered by cropland, (ii) a fully afforested world with extensive wood harvesting, and (iv) a full cropland world with extensive irrigation. In these idealized sensitivity experiments, performed under present-day climate conditions, the effects of the different LCLMC strategies represent an upper bound for the potential of global mitigation and

local adaptation. To disentangle the local and non-local effects from the LCLMC, a checkerboard-like LCLMC perturbation, i.e., alternating grid boxes with and without LCLMC, is applied. The local effects of deforestation on surface temperature are largely consistent across the ESMs and the observations, with a cooling in boreal latitudes and a warming in the tropics. However, the energy balance components driving the change in surface temperature show less consistency across the ESMs and the observations. Additionally, some biases exist in specific ESMs, such as a strong albedo response in CESM mid-latitudes and a

soil thawing driven warming in boreal latitudes in EC-EARTH. The non-local effects on surface temperature are broadly consistent across ESMs for afforestation, though larger model uncertainty exists for cropland expansion. Irrigation clearly induces a cooling effect, however; the ESMs disagree whether these are mainly local or non-local effects. Wood harvesting is found to





have no discernible biogeophysical effects on climate. Our results overall underline the potential of ensemble simulations to inform decision making regarding future climate consequences of land-based mitigation and adaptation strategies.

## 1 Introduction

Land cover change and land management change have been intrinsically connected to human development throughout history. The impact of land cover change and land management change (LCLMC) on the global carbon cycle was estimated at 116 PgC based on global compilations of carbon stocks for soils (Sanderman et al., 2017) and for vegetation as 447 PgC (Erb et al. (2018): a loss of about half of the world's terrestrial biomass), with substantial shares already in the pre-industrial pe-

riod (Canadell et al., 2021). About 10% of anthropogenic CO2 emissions have been caused by LCLMC over the last decade (Friedlingstein et al., 2022). According to integrated assessment models, LCLMC will play an important role in the near-term future as most low-end warming scenarios assume large-scale deployment of land-based mitigation (IPCC, 2018). However, the effect of changed land cover and management on the climate is still highly uncertain and poorly understood (Mahmood et al., 2014; Pitman et al., 2009; Perugini et al., 2017). For instance, land use policies generally only account for the effects on

the carbon balance while largely neglecting the biogeophysical effects (Duveiller et al., 2020). These biogeophysical effects include (i) the effects of land cover and land management change on the surface radiation budget (e.g. a forest is a darker surface than open grass or cropland hence it absorbs more shortwave radiation) (ii) the effects of non-radiative processes like changes in evaporative efficiency and surface roughness, and (iii) the effects induced by atmospheric circulation through altering heat, moisture and momentum transport (Bright et al., 2017; Winckler et al., 2017; Duveiller et al., 2020). The induced

changes in atmospheric circulation are often classified as non-local processes as they typically affect other regions than those where the LCLMC occurred. The effects on surface radiation and surface properties are called local processes, as they are a direct consequence of local LCLMC.

As LCLMC are an often cited approach for local mitigation and adaptation policies (Minx et al., 2018; Perugini et al.,

2017), the separation of local and non-local effects can help in reducing current uncertainty in assessments of biogeophysical effects. As non-local effects are a consequence of LCLMC occurring elsewhere, they are generally not a desired effect from specific policies (which tend to have a local scope), but rather an undesired and unintended effect from LCLMC across the globe. In contrast, local effects from LCLMC are directly influenced by local decisions and can be applied more directly in local adaptation and/or mitigation policies. Therefore, the separation between local and non-local effects is beneficial for the

implementation of biogeophysical effects related to LCLMC in local mitigation and adaptation policies.

A first set of studies attempted to use Earth System Models (ESMs) to understand the global effects of land cover change, both in idealised (Davin and de Noblet-Ducoudre, 2010; Boysen et al., 2020; Meier et al., 2021) and in more realistic setups (Pitman et al., 2009; Pongratz et al., 2010; Boisier et al., 2012; Ito et al., 2020). However, these studies only show aggregated

effects of the biogeophysical processes highlighted above and no direct separation is made between effects caused by local and



non-local processes. Some studies extracted the local signals from Earth System Model (ESM) simulations by comparing data at tile level (Malyshev et al., 2015) or extracting local signals by comparing neighbouring grid cells with different land cover change rates (Kumar et al., 2013; Lejeune et al., 2017). Nevertheless, these approaches either have a limited spatial coverage (Kumar et al., 2013; Lejeune et al., 2017) or are limited to ESMs with tile level output data (Malyshev et al., 2015). A novel

approach by Winckler et al. (2017), often referred to as the checkerboard approach, separates land cover change signals into local and non-local effects without these limitations. This was done by prescribing a land cover map with grid cells which underwent land cover change and grid cells with the original land cover in a regular pattern (e.g. 1/8, 1/4, etc.). By contrasting this simulation to a reference simulation without land cover change, the local and non-local signals can be separated. However, the simulations performed in Winckler et al. (2017) are limited to a single ESM (MPI-ESM, Winckler et al. (2017, 2019b, a, c)).

Multi-model studies, like the step-wise deforestation experiment within the Land Use Model Intercomparison Project (LUMIP, Boysen et al. (2020)) report local and non-local effects by comparing results within and beyond the geographical region of deforestation, which, however does not allow for a quantitative separation on the global scale.

A second set of studies investigated the climate effects of land cover change based on observational data. Remote sensing

data is used to compare the surface temperature of a forested patch and a patch of open land, both spatially (Duveiller et al., 2018; Li et al., 2015) and temporally (Alkama and Cescatti, 2016). Data from eddy covariance towers providing direct flux measurements (e.g. through FLUXNET) were used to reconstruct the biogeophysical effects of deforestation (Bright et al., 2017). These observational estimates by design exclude the non-local signals which might dominate the response to deforestation, according to recent work applying the alternating LCLMC approach (Winckler et al., 2019a).


Unlike land cover change, the climate effects of land management change, like irrigation and wood harvesting, are less studied. This is remarkable, as observational studies indicate that both land cover change and land management change have an equally important effect on climate variables such as surface temperature (Luyssaert et al., 2014). Moreover land management will be increasingly important towards the future due to land scarcity and the need for intensification as well as the additional

pressure on land for carbon dioxide removal (Pongratz et al., 2021). Among land management change options, irrigation has a clear regional cooling effect, especially during warm episodes (Hirsch et al., 2017; Thiery et al., 2017, 2020; Chen and Dirmeyer, 2019; Gormley-Gallagher et al., 2020; Mishra et al., 2020). Despite its recognised imprint on local climate, only a few ESMs simulate irrigation explicitly, with only three ESMs including irrigation in the CMIP6 simulations (Al-Yaari et al., 2022). Wood harvesting has mostly been studied for its biogeochemical effects while the analysis of the biogeophysical effects

is still lacking in studies using ESMs. Observational studies, however, indicate an effect of wood harvesting on albedo (Otto et al., 2014) and surface roughness (Nakai et al., 2008). Furthermore, the effect of land management change on atmospheric circulation has been hypothesised, with for instance irrigation induced cooling causing a delayed onset of the Indian Monsoon (Guimberteau et al., 2012; Thiery et al., 2017) and modified precipitation patterns in Eastern Africa (De Vrese et al., 2016). Yet, the relative importance of local versus non-local effects induced by land management changes has not been studied so far.




In this study, we quantify the sensitivity of local and non-local climate to LCLMC and investigate the processes contributing to surface temperature changes. We apply the checkerboard approach to idealised simulations in a multi-model framework using three ESMs. Idealised simulations are performed with two land cover change sensitivity experiments (cropland expansion and afforestation), and two land management change sensitivity experiments (irrigation and wood harvest expansion). The simulations represent changes from present-day land cover, and thus provide policy makers with information on potential effects of LCLMC under present-day climate. First, we describe the spatial patterns of the local and non-local effects of surface temperature to the LCLMC sensitivity experiments. Second, we evaluate the local effect in the different ESMs for deforestation against estimates derived from observations and remote sensing (Duveiller et al., 2018; Alkama and Cescatti, 2016; Li et al., 2015; Bright et al., 2017). Finally, we analyse the processes underpinning the local effect of different LCLMC using an energy balance decomposition.

## 2 Methods

### 2.1 ESM sensitivity experiments

#### 2.1.1 Participating ESMs

Three state-of-the-art ESMs are used in this study: the Community Earth System Model (CESM), the Max Planck Institute for Meteorology Earth System Model (MPI-ESM), and the European Consortium Earth System Model (EC-EARTH). Here, we provide a brief technical description of each model.

We use CESM version 2.1.3 (hereafter referred to as CESM), an open-source and fully coupled ESM (Danabasoglu et al., 2020). CESM combines the Community Atmosphere Model version 6 (CAM6), the Community Land Model version 5 (CLM5; Lawrence et al., 2019), the Parallel Ocean Program version 2 (POP2), The Community Ice Sheet Model (CISM), the Los Alamos National Laboratory Sea Ice model (CICE), and the Model for Scale Adaptive River Transport (MOSART). CESM has some notable improvements to the previous version (Danabasoglu et al., 2020); for instance, CLM5 includes improvements in the snow and plant hydrology, the lake model, and carbon and nutrient recycling (Lawrence et al., 2019). CLM5 also includes 14 natural Plant Functional Types (PFTs) and 8 Crop Functional Type (CFTs), whereby CFTs can exist either on a rainfed patch or an irrigated patch. The Sub-grid heterogeneity is implemented using a nested hierarchy where an individual grid cell constitutes of different land units such as vegetated, urban, lake, glacier and crop fractions (Lawrence et al., 2019). The CESM simulations were performed at a spatial resolution of 0.90°x1.25°.

The Max Planck Institute for Meteorology Earth System Model version 1.2 with low resolution configuration (MPI-ESM1.2-LR; hereafter referred to as MPI-ESM) is a fully coupled state-of-the-art ESM that uses the atmospheric component ECHAM6.3 and the land component JSBACH3.2 (around 200 km horizontal resolution (T63) and 47 atmospheric vertical levels), which



are coupled via OASIS3-MCT to the ocean dynamic (MPIOM1.6) and ocean biogeochemistry (HAMOCC6) models (around 150 km grid spacing and 40 vertical levels). A detailed description of MPI-ESM1.2 can be found in (Mauritsen et al., 2019).

A similar setup has been also used within CMIP6/LUMIP, e.g. with studies on biogeophysical effects of deforestation (Boysen et al., 2020) as well as other recent studies on the effects of land use and land cover change on climate (Winckler et al., 2019a, b). JSBACH3.2 simulates in total 12 different plant functional types (PFTs), with 4 forest PFTs (tropical broadleaf evergreen and deciduous trees, extra-tropical evergreen and deciduous trees) and two cropland PFTs (C3 and C4 crops). The MPI-ESM simulations were performed at a spatial resolution of 1.88°x1.88°.


EC-EARTH is a state-of-the-art Earth system model developed by the EC-Earth consortium (Döscher et al., 2021). In this study we use the released version EC-Earth3-Veg (v3.3.3.1). The atmospheric component is the Integrated Forecast System (IFS) developed by the European Centre for Medium Range Weather Forecasts (ECMWF) that uses the TL255 horizontal grid (+-80 km) and 91 vertical model levels with the top level at 0.01 hPa. The oceanic component is the Nucleus for European

Modelling of the Ocean (NEMO) model (v3.6). The vegetation model is the Lund-Potsdam-Jena General Ecosystem Simulator (LPJ-GUESS). Note that this is a dynamic vegetation model which does not explicitly solve the energy balance as the previous ESMs did. The atmosphere model IFS has a dedicated land surface component : the Hydrology Tiled ECMWF Scheme for Surface Exchanges over Land (HTESSEL) to handle the surface water and energy fluxes to the atmosphere. In LPJ-GUESS, the vegetation dynamics for the land are simulated on 6 stand types, namely Natural, Pasture, Urban, Crop, Irrigated Crop and

Peatland. In the Natural stand 10 woody and 2 herbaceous PFTs compete (Smith et al., 2014). On Pasture, Urban and Peatland fractions 2 herbaceous species are simulated, conforming to the C3 and C4 photosynthetic pathways. The Crop stands have 5 CFTs, both annual and perennial C3 and C4 crops, and C3 N fixers (Lindeskog et al., 2013). The EC-EARTH simulations were performed at a spatial resolution of 0.7°x0.7°.

There are some important differences in how the different ESMs treat land cover. They have a different amount of PFTs which are also defined in different categories. Moreover while in MPI-ESM and CESM land cover is handled within one single sub-model (their respective land surface schemes JSBACH and CLM) and is prescribed, in EC-EARTH there are different models for vegetation dynamics and biogeochemistry (LPJ-GUESS) and for the water and energy cycle (HTESSEL). We summarize the most important differences relating to how the ESMs handle land cover in table 1. Additionally, in order to give

an idea of the differences in the initial land cover maps, we provide the 2015 forest fractions (evergreen, deciduous and total forest) for all ESMs in Appendix A.

### 2.1.2 Experimental design

We conducted four idealised LCLMC simulations and one reference simulation using the three ESMs. Every simulation has the same set-up, but differs in terms of land cover and land management. As we want to remain independent of any future

climate scenarios, the simulations will be performed under present-day (2015) climate forcing. They will cover the entire globe as to inform on where LCLMC might be more or less useful. Four idealised sensitivity experiments are investigated: (i) a



**Table 1.** Specifications of how land cover is handled across the different ESMs.

| ESM | Land Model | spatial resolution | amount of PFTs | Prescribed land cover |
|---|---|---|---|---|
| CESM2 | CLM5 | 0.90°x1.25° | 14 | yes |
| MPI-ESM1.2 | JSBACH3.2 | 1.88°x1.88° | 12 | yes |
| EC-EARTH3-Veg | LPJ-GUESS/HTESSEL | 0.7°x0.7° | 12 | no |

fully afforested world (FRST), (ii) a full cropland world (CROP), (iii) a fully afforested world with extensive wood harvesting (HARV), and (iv) a full cropland world with extensive irrigation (IRR). In order to be able to distinguish between the local and non-local effects of these four idealised cases, the LCLMC perturbations are applied following the checkerboard approach of

Winckler et al. (2017) using a checkerboard pattern which is detailed in section 2.1.3, effectively meaning that only half of the grid cells undergo LCLMC. In addition, a control simulation with present-day land cover is performed by every ESM to serve as a reference (hereafter referred to as CTL). The CTL simulation uses the native, present-day land cover map of each ESM, which are all based on the Land Use Harmonization version 2 dataset (LUH2; Hurtt et al., 2020). This implies that each ESM retains its native PFTs. The CTL simulation does not include land management (i.e. irrigation and wood harvesting are set to

zero) to have a clear baseline for the sensitivity simulations. In all simulations, anthropogenic forcing (including greenhouse gas and aerosol concentrations) is kept constant at the 2015 conditions. The initial conditions are provided by the CMIP6 historical simulations in 2014 and applied to the different ESMs to conduct model simulations for a period of 160 years. The first 10 years are considered as biogeophysical spin-up and omitted in the analysis. We let the stratospheric aerosols evolve transiently until 2025 based on data from the Scenario Model Intercomparison Project (ScenarioMIP), after which they are

kept constant. This was done to ensure that the stratospheric aerosol concentrations in our simulations resemble the mean state of the 21th century. Due to technical constraints in CESM however, the 2025 levels were used from the start of the simulation. The solar forcing is kept at natural oscillations, except for CESM where these are set to a constant value that is chosen equal to the average over the entire simulation period. This is needed to ensure that all ESMs have the same amount of solar energy entering the system over the entire simulation period. Overall, the set-up is designed to represent present-day climatic conditions

through model simulations sufficiently long to average out internal variability. All simulations are performed in fully coupled mode, consistent with the LUMIP protocol (Lawrence et al., 2016), and at each ESM's typical spatial resolution employed for CMIP6 (lat x lon) (MPI-ESM:1.88°x1.88°, CESM:0.90°x1.25°, EC-EARTH:0.7°x0.7°).

The different LCLMC scenarios used in the sensitivity experiments are outlined in Table 2. The idealised land cover maps

for CESM and MPI-ESM are constructed following the approach described in Davin et al. (2020) using prescribed idealised land cover maps. To create the idealised FRST land cover map, we start from the 2015 land cover map of each model. All PFTs that are neither forest nor bare soil were removed. The remaining forest fractions are increased such that fractions within a grid cell add up to 100 %. As the bare soil fraction is preserved, the resulting land cover map only contains forest PFTs and bare



**Table 2.** Overview of simulation set-up for the ESMs.

| Simulation | Land Cover | Land Management |
|------------|------------|-----------------|
| CTL | 2015 map | none |
| CROP | 50% crop map, 50% CTL map | none |
| FRST | 50% forest map, 50% CTL map | none |
| IRR | 50% crop map, 50% CTL map | irrigation |
| HARV | 50% forest map, 50% CTL map | wood harvesting (yr 2100 under RCP 8.5) |

soil. The approach mimics forest expansion across all vegetated, cropland and urban areas but avoids that trees are planted in
e.g. desert, high altitude, and tundra regions (Figure 1d-f). Note that this approach is only possible for grid cells containing
forest PFTs. For grid cells without forest PFTs present, we calculate the latitudinal average (at each ESMs native resolution)
of the relative forest PFT distribution consisting of different species. This value is then considered as representative for this
latitudinal band and is used to replace all other vegetation in the grid cell. The same approach is followed for constructing the
CROP map by keeping the crop fraction constant within a grid cell and removing all non-cropland PFTs (e.g. pasture, bush,
forest and grassland; Figure 1a-c).

The same approach was not entirely possible in EC-EARTH as this version of the model has a dynamic vegetation model
(Döscher et al., 2021). Therefore, to obtain a simulation that is as close as possible to a 100% forest world, the managed veg-
etation is turned off. Consequently, the fully forested world simulation in EC-EARTH can also contain grasses. For the CROP
simulation, the natural land cover is switched off, which forces the model to only grow crops within a grid cell. As in the other
ESMs, bare soil fractions were retained, while only vegetated areas and urban areas where considered for land cover change.
Note that this difference in implementation of the LCLMC has led to strong differences in the total extent of the LCLMC, most
notably regarding the afforestation experiment where EC-EARTH shows little afforestation in contrast to MPI-ESM and CESM
(Figure 1f). These low amounts of afforestation modelled in the EC-EARTH FRST simulation make that it is less comparable
to the other ESMs for this land cover change.

For the IRR simulation, we apply the same land cover maps as in the CROP simulation, but here, the native irrigation param-
eterisation of each model is activated and applied at the global scale (Figure 1g-i). Although the individual implementations
of the irrigation parameterisation differ, all models follow a similar logic. Once a crop suffers a certain amount of water stress
(defined differently in the models, see Appendix B), this amount is replenished by applying an irrigation flux until the water
stress is relieved. In CESM and EC-EARTH, no limit is imposed on water available for irrigation. In MPI-ESM however, water
availability is limited by the amount of runoff and drainage in the grid cell.



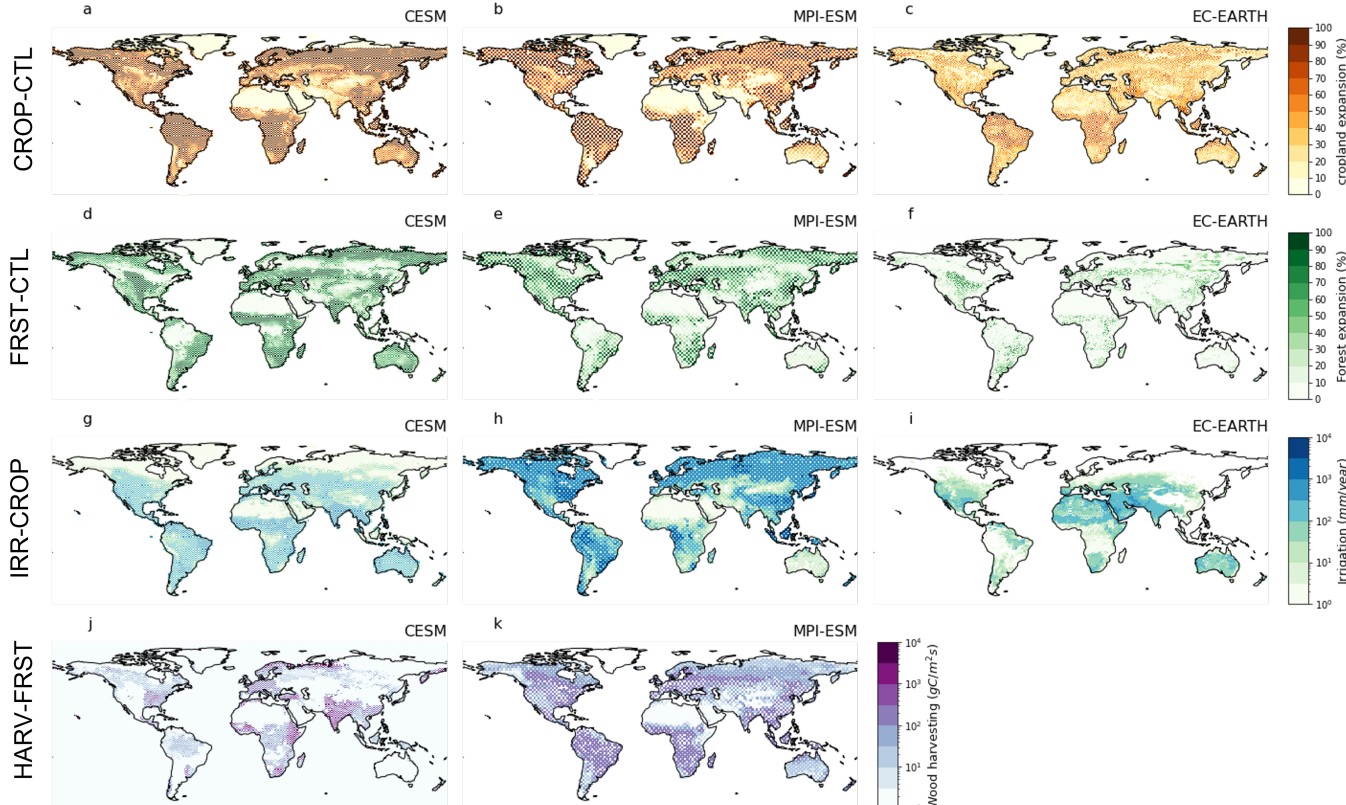

**Figure 1.** Overview of land cover and management changes modelled in the ESM sensitivity experiments. The amount of cropland expansion is shown for the CROP simulation as compared to present-day land cover (CTL) for CESM (a), MPI-ESM (b) and EC-EARTH (c). The amount of afforestation in the FRST simulation as compared to present day land cover (CTL) is shown for CESM (d), MPI-ESM (e) and EC-EARTH (f). Both land cover changes are shown as an area fraction of the land cover in that grid cell. The amount of wood harvest applied in the HARV simulation as compared to the FRST simulation is shown for CESM (g) and MPI-ESM (h) in terms of intensity of harvesting (gC m$^{-2}$s$^{-1}$). Finally the amount of irrigation is shown as expressed in a discharge (mm year$^{-1}$) for CESM (i), MPI-ESM (j) and EC-EARTH (k). Do note that the color bar is exponential for land management change (g-k) while it is linear for land cover change (a-f).

The amount of wood harvesting is typically a prescribed value in ESMs, often expressed as an amount of biomass carbon
extracted from the PFTs. In the HARV simulation, we force the models to use the wood harvest rates specified in the CMIP6
SSP5-8.5 scenario by the end of the century (Figure 1j-k). We let the forest grow as in the FRST simulation without harvesting
for the first 40 years to build up biomass before prescribing the intensive wood harvest rates. For the remaining 120 years of
the simulation, the harvest rates are kept constant. It should be noted that EC-EARTH did not provide this simulation. In MPI-
ESM, there is no feedback implemented of this management practice to any atmospheric processes. Therefore, only CESM
can be used to investigate the biogeophysical effects due to wood harvesting.



The idealized sensitivity experiments are conducted under present-day climate forcing. The effects of the different LCLMC strategies represent an upper bound on the potential for global mitigation and local adaptation against the current background climate. They should therefore not be perceived as realistic futures. Both CESM and MPI-ESM show extreme land cover changes in the CROP and FRST simulations compared to CTL (Figure 1a-f). Overall, the land cover change is stronger in CESM than in MPI-ESM, but the spatial patterns roughly match. Some notable differences include the extent of cropland expansion in Siberia and the amount of afforestation in Australia. Do note that in panels (a) and (b), the amount of cropland expansion (i.e. all conversions to crop) shown is not equivalent to the amount of deforestation (i.e. all conversion from forest to crop) in these simulations as other conversions (e.g. bush and grassland to crop) also occur.

The comparison of land management between CESM and MPI-ESM shows strong differences, despite using a qualitatively consistent implementation across both ESMs. For wood harvesting, the spatial pattern and intensity differ notably. In CESM the wood harvesting is generally more intense locally and less homogeneous across space than in MPI-ESM (Figure 1j-k). For irrigation the spatial extent also differs strongly between the models. Most notably, due to the simple irrigation scheme implemented in MPI-ESM (see appendix B), this model shows high irrigation amounts in the boreal latitudes while there is no irrigation occurring in CESM and EC-EARTH at these latitudes (Figure 1g-i).

### 2.1.3 Extraction of local and non-local signals

To disentangle the local and non-local effects due to LCLMC, the checkerboard approach of Winckler et al. (2017) is applied, which is described here briefly (see Winckler et al. (2017) for details). The checkerboard approach alternates LCLMC grid cells with grid cells which remain unaltered. This allows for a clean separation of local and non-local effects as the latter only occur over unaltered grid cells while the grid cells where LCLMC did occur represent a combination of both local and non-local effects. In our simulations, 1 out of 2 grid cells are affected by the LCLMC and these cells are spread out in a regular checkerboard pattern. The checkerboard like LCLMC alternation is applied to all simulations except the CTL simulation. This means that for each simulation, only half of the grid cells undergo LCLMC. The remaining unchanged grid cells show the exact same land cover as the CTL simulation. The 150 year-simulation is split into 5 slices of 30 years each. To account for natural variability, we treat each slice as a member of a perturbed initial condition ensemble. A multi-year monthly mean is computed over each of these ensemble members. To extract the local and non-local signals, we subtract a land cover change member (CROP, FRST) from its corresponding CTL member. The resulting signals for grid cells where no land cover change occurred cannot be ascribed to any direct (i.e. local) land cover change effect and can therefore be ascribed entirely to non-local effects caused by LCLMC in other grid cells. We then spatially interpolate (using linear interpolation) these values to get a global map of non-local effects. The differences between both ensemble members for grid cells where land cover change did occur are caused by both local and non-local effects (local effects stem from the land cover change within the grid cell, while non-local effects are caused by land cover change in other grid cells). Hence, these non-local effects are subtracted from the total combined effect to get a local signal. As this local signal can only be calculated over the grid cells where land cover change



**Table 3.** Overview of observational products available for the different variables considered in the evaluation.(*) This data was first published in 2018 but later extended to cover a larger area in 2020, as the extended dataset is used in this study,we will refer to this dataset as DV20 from hereon. (**) Note that the sensible heat flux was obtained by the closure of the energy balance.

| Dataset | Data Type | Available variables |
| --- | --- | --- |
| Duveiller et al. (2018, 2020)* | remote sensing | surface temperature, latent heat flux, sensible heat flux**, albedo |
| Duveiller et al. (2021) | remote sensing | near-surface air temperature |
| Li et al. (2015) | remote sensing | surface temperature, latent heat flux, albedo |
| Alkama and Cescatti (2016) | remote sensing | surface temperature, near-surface air temperature |
| Bright et al. (2017) | flux towers | surface temperature |

occurred, we again spatially interpolate this pattern to get a full global map. Finally, the local and non-local signals are summed up to derive the total signal, which corresponds to the signal from an idealised global experiment without the checkerboard-like LCLMC pattern applied. The checkerboard approach is implemented to each model grid at its native resolution. Hence, grid cell sizes vary across the different ESMs. As we have five ensemble members of 30 years for each simulation, we can extract

local and non-local signals for each ensemble member, which are then used as a measure of uncertainty coming from natural variability.

The procedure described above can be extended to land management change by using one of the land cover change simulations as a reference simulation instead of the CTL simulation. To extract the signal from irrigation expansion, the IRR

simulation is compared against the CROP simulation. In case of wood harvesting, the HARV simulation is compared to the FRST simulation.

## 2.2    Evaluation of local signal to deforestation

The modeled responses induced by deforestation are evaluated against products from observational studies. Several studies

provide global estimates of the effect of deforestation with remote sensing products (Li et al., 2015; Alkama and Cescatti, 2016; Duveiller et al., 2018) or ground observations (FLUXNET Bright et al., 2017). Only the local signals can be compared here as these observations only capture local effects by design (Winckler et al., 2019b). All four observational studies represent an idealised case where a fully deforested patch of land is compared to a fully forested patch. We therefore use the local signals derived from comparing the CROP to the FRST simulation to evaluate the ESM response to deforestation against these

products. It was shown by Winckler et al. (2019b, c) that a comparison between modelled response and these observational estimates is useful to evaluate the performance of ESMs to represent the effects of LCLMC on surface temperature.



The evaluation is also performed for several other variables of interest, including latent heat flux, sensible heat flux, albedo and near-surface air temperature (2 m temperature, tas in CMIP6 nomenclature), however not all of these are available in each

dataset (see Table 3). The spatial extent of the observational studies varies strongly, therefore the evaluation will be performed along latitudinal bands following Meier et al. (2018) to focus on the global patterns. A description of the different observational datasets used and their spatial maps are provided in appendix C.

## 2.3 Energy balance decomposition for changes in surface temperature

An energy balance decomposition approach is used to decompose the change in surface temperature to its driving surface processes. Here, we use this approach to understand the processes underlying the modelled effects of LCLMC. We use the approach developed by Juang et al. (2007) and modified by Luyssaert et al. (2014) which has often been used in LCLM studies, notably with CLM (Akkermans et al., 2014; Hirsch et al., 2018; Thiery et al., 2017; Hauser et al., 2019; Vanderkelen et al., 2021). The energy balance equation is shown below.

$$\epsilon \sigma T_s^4 = (1-\alpha)SW_{in} + LW_{in} + LHF + SHF \tag{1}$$

Where $\epsilon$ is the surface emissivity, $\sigma$ is the Stefan-Boltzmann constant ($5.67 \times 10^{-8}$ W m$^{-2}$ K$^{-4}$), $T_s$ is the radiative surface temperature as it is directly calculated from surface upwelling longwave radiation, $\alpha$ is the surface albedo, and $SW_{in}$ and $LW_{in}$ are the incoming shortwave and incoming longwave solar radiation, respectively. $LHF$ and $SHF$ are the latent and sensible heat flux, respectively. All fluxes are expressed in W m$^{-2}$. We take the total derivative to obtain the change in surface

temperature, whereby $\epsilon$ can be assumed to be equal to 1 for the application of this equation (Juang et al., 2007; Luyssaert et al., 2014).

$$\Delta T_s = \frac{1}{4\sigma T_s^3}(-SW_{in}\Delta\alpha + (1-\alpha)\Delta SW_{in} + \Delta LW_{in} - \Delta LHF - \Delta SHF) \tag{2}$$

Here, we apply the energy balance decomposition only to the local effects derived from the LCLMC signals as these are directly linked to changes in surface properties (Winckler et al., 2017). While applying this approach, a modest global imbalance

of less than 0.1 W m$^{-2}$ is found over all land grid cells for all different cases, indicating the general applicability of the method.

## 3 Results

### 3.1 Evaluation of biogeophysical response to deforestation

We compare observational estimates to the simulated local response of full deforestation (CROP-FRST, i.e. the idealised effect of going from a fully forested to a fully cropland world), in order to evaluate the modelled response to deforestation of the




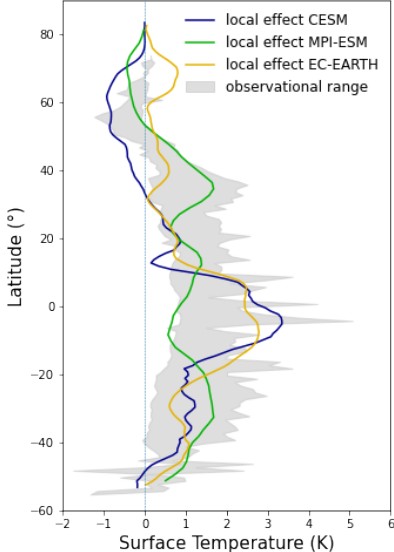

**Figure 2.** Latitudinal evaluation of local surface temperature derived from full deforestation experiments (CROP-FRST) for CESM (blue), MPI-ESM (green) and EC-EARTH (yellow). Note that for all ESMs a running latitudinal mean of 2° was computed. The observational range (grey shade) shows the full range given by four observational estimates (Li et al., 2015; Alkama and Cescatti, 2016; Duveiller et al., 2018; Bright et al., 2017).

different ESMs. The latitudinal response of the average annual local surface temperature for all ESMs are generally within the observational range (Figure 2). The latitudinal change in surface temperature is similar to the observational estimates: a warming in the tropics (up to 3 K) and a cooling in the Northern Hemisphere (NH) boreal latitudes (up to -1 K). Only EC-EARTH deviates from this as it shows no cooling in NH boreal latitudes (50°N-80°N) and even shows a warming. CESM simulates a different sign compared to observations in the NH mid-latitudes (30°N-50°N), but performs reasonably well at boreal latitudes.

Overall, MPI-ESM matches reasonably well to the observational estimates. In the tropics, MPI-ESM simulates values near the lower bound of the observational range (0.6 K), while CESM and EC-EARTH simulate values near the upper bound (3 K). In general, all models show a reasonable agreement with the observations, both in sign and magnitude over most latitudes; only in the NH mid-latitudes and boreal latitudes the models diverge from the observed range.

Comparing the local effect of deforestation on surface temperature across seasons generally shows a good agreement of MPI-ESM with the observational estimates for the different seasons (Figure 3). The CESM simulations lie within the observational range for boreal winter and fall but show a cooling response to deforestation in boreal spring and summer above 30°N, which is in contrast to the observed warming. The EC-EARTH simulations agree well with the observations except for the boreal latitudes where a sustained warming occurs over all seasons except during the boreal summer.






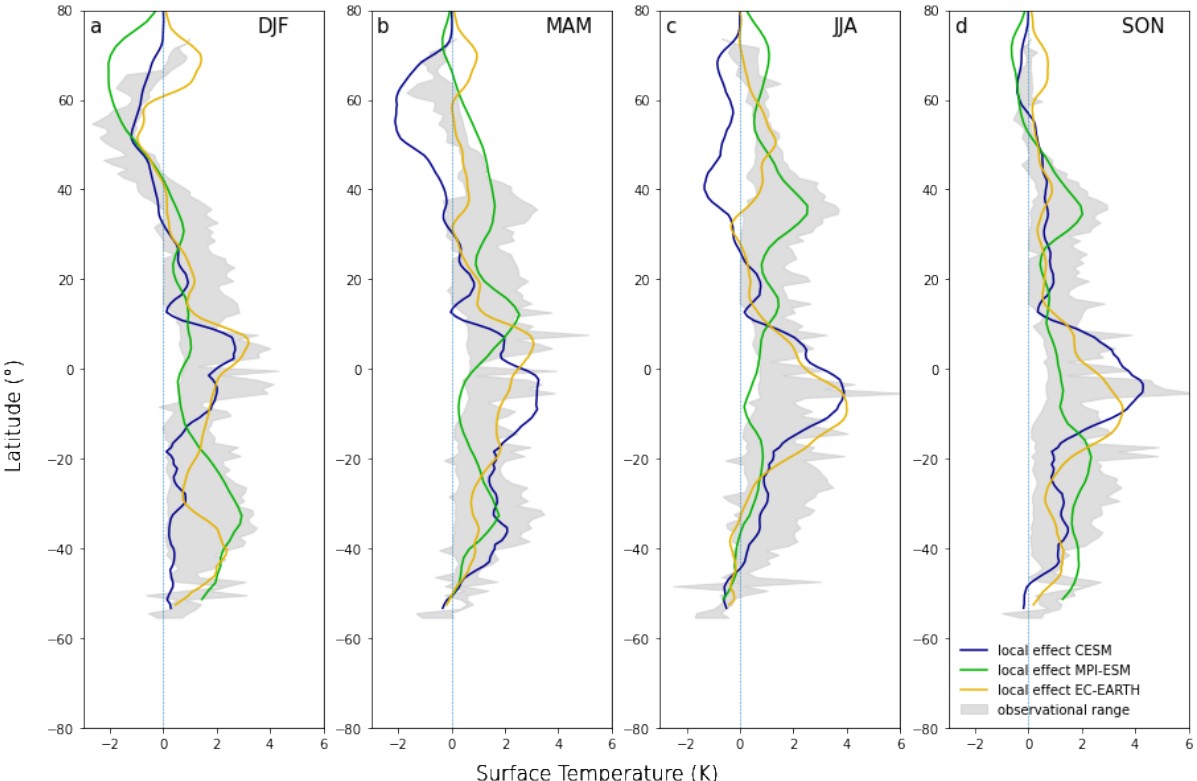

**Figure 3.** Latitudinal evaluation of local surface temperature derived from full deforestation experiments (CROP-FRST) for CESM (blue), MPI-ESM (green) and EC-EARTH (yellow) for different seasons, winter or DJF (December, January, February) in panel a, spring or MAM (March, April, May) in panel b, summer or (June, July, August) in panel c and fall or (September, October, November) in panel d. Note that for all ESMs a running latitudinal mean of 2° was computed. The observational range (grey shade) shows the full range of values spanned by four observational estimates (Li et al., 2015; Alkama and Cescatti, 2016; Duveiller et al., 2018; Bright et al., 2017).

The effect of deforestation on annual local latent and sensible heat fluxes agrees well with the observational estimates for all ESMs (Figure 4a, b). The latent heat flux is modelled to decrease over most latitudes. MPI-ESM underestimates the magnitude of the latent heat flux signal over most of the subtropics, and shows an overestimation over the boreal latitudes. CESM and EC-EARTH match well to the observations, except at the mid-latitudes where it underestimates the decrease in latent heat flux.
EC-EARTH shows no change in latent heat flux except over the tropics where a clear decrease is shown.

Observations show a deforestation-induced decrease in sensible heat flux in the extra-tropics, a slight increase around 20°N and 20°S and a decrease around the Equator. CESM captures the response in sensible heat flux well in the NH but overestimates it in the tropics and projects an opposite sign over most of the Southern Hemisphere (SH). MPI-ESM underestimates the
change over most of the latitudes and shows an increase instead of a decrease at boreal latitudes. Similar to the latent heat flux,



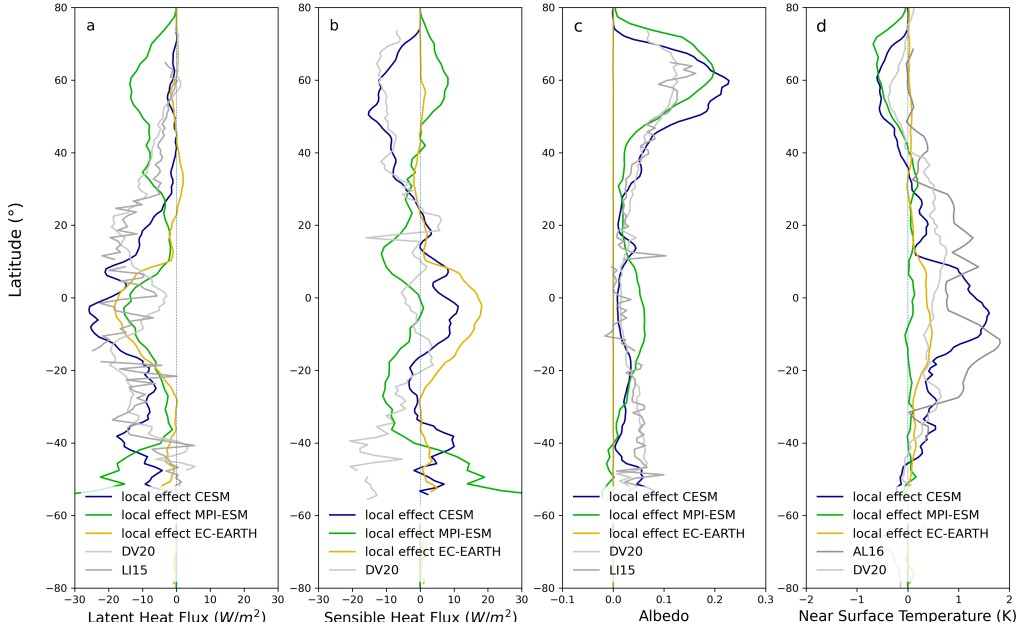

**Figure 4.** Latitudinal evaluation of local energy and climate variables derived from full deforestation experiments (CROP-FRST). The local effect simulated by CESM (blue), MPI-ESM (green) and EC-EARTH (yellow) of latent heat flux (W/m$^2$ ) (a) compared to observational estimates by Li et al. (2015); Duveiller et al. (2018) (DV20 and LI15, respectively), of sensible heat flux (W m$^{-2}$) (b) compared to Duveiller et al. (2018) (DV20), of albedo (-) (c) compared to Li et al. (2015); Duveiller et al. (2018) (LI15 and DV20) and near surface temperature (K) (d) compared to Alkama and Cescatti (2016); Duveiller et al. (2020) (AL16 and DV20). Note that for all ESMs a running latitudinal mean of 2° was computed.

EC-EARTH only shows a non-zero effect over the tropics where the model suggests a strong increase. These strong biases in both latent and sensible heat fluxes of MPI-ESM and EC-EARTH do not appear to affect the surface temperature responses. This could partially be explained by opposite signs in the biases of both turbulent heat fluxes, which cancel each other out, as is likely the case over boreal latitudes for MPI-ESM and in the tropics for EC-EARTH.


The deforestation-induced albedo change is especially important at boreal latitudes where it dominates the overall surface temperature response (Davin and de Noblet-Ducoudre, 2010). CESM captures the observed albedo response well, except north of 40°N where it overestimates the albedo change and south of 30°S where it underestimates the albedo change (Figure 4c). MPI-ESM shows a similar bias in the SH. It also overestimates the brightening in the tropics and boreal latitudes following 330 deforestation and underestimates the brightening over most mid-latitudes.





The bias in albedo response north of 40°N could be caused by a strong snow masking response in both ESMs, as a snow covered forest is darker than a snow covered cropland. This would also explain the strong cooling in boreal spring and summer seasons in CESM (Figure 3b,c) and the bias in annual surface temperature over the mid-latitudes (Figure 2). In EC-EARTH the local albedo change is zero (Figure 4c), however there is a stronger non-local albedo change despite this being almost absent in other ESMs (Figure D1). The non-local albedo change is near-zero except over boreal latitudes, where it agrees in sign with observations but strongly underestimates the magnitude (Figure D2). The results for CESM are in contrast to Meier et al. (2018) who showed that the previous version of CLM (CLM4.5) could reproduce the observed albedo relatively well. However the differences between our results might be due to differences in model setup as CLM was evaluated in offline mode in Meier et al. (2018) in contrast to the coupled simulations performed here.

The near-surface air temperature is often a preferred metric compared to the surface temperature, as it is more relevant for understanding the perceived temperature and is considered in most policy-relevant metrics including those used to measure global warming (Arias et al., 2021). For local near-surface air temperature change, CESM and EC-EARTH show a response of similar sign to the observations in the SH and tropics. The observations diverge north of 40°N, where the DV20 dataset confirms the cooling which is simulated by CESM and MPI-ESM. In contrast, the AL16 dataset shows no temperature change, which is also the case for EC-EARTH (Figure 4d). The near-surface air temperature in MPI-ESM is relatively insensitive to deforestation except north of 40°N as was also shown in (Winckler et al., 2019c). However, it should be considered that near-surface air temperature is a highly contested measure as its definition tends to vary strongly across different ESMs, especially over grid cells or grid cell fractions covered with tall vegetation (Boysen et al., 2020; Winckler et al., 2019c). Therefore, in the remainder of this study, we will focus on the response of LCLMC on surface temperature, while the maps for near-surface air temperature are added in appendix D for reference.

## 3.2 Local and non-local effects of LCLMC on surface temperature

This section provides an overview of the signal separated effects on surface temperature of the different LCLMC across the different ESMs. We discuss the local, non-local and total effects per LCLMC category. At the end of the section the changes which are consistent across all ESMs are summarized in Table 4.

### 3.2.1 Cropland expansion

As a consequence of cropland expansion (CROP-CTL), CESM shows a strong local cooling over the NH boreal latitudes which extends into most of the NH mid-latitudes (Figure 5a). The tropics and subtropics show a strong local warming of up to 4 K over the (deforested) tropical rainforests. MPI-ESM shows a similar pattern to CESM in NH boreal latitudes but with a smaller local cooling which does not extend as far south into the NH mid-latitudes. MPI-ESM also simulates local warming over the tropics, but with a different spatial pattern and magnitude compared to CESM and EC-EARTH. The local signals in EC-EARTH are similar to CESM, showing a strong local warming in the tropics. However, in NH boreal latitudes the signals





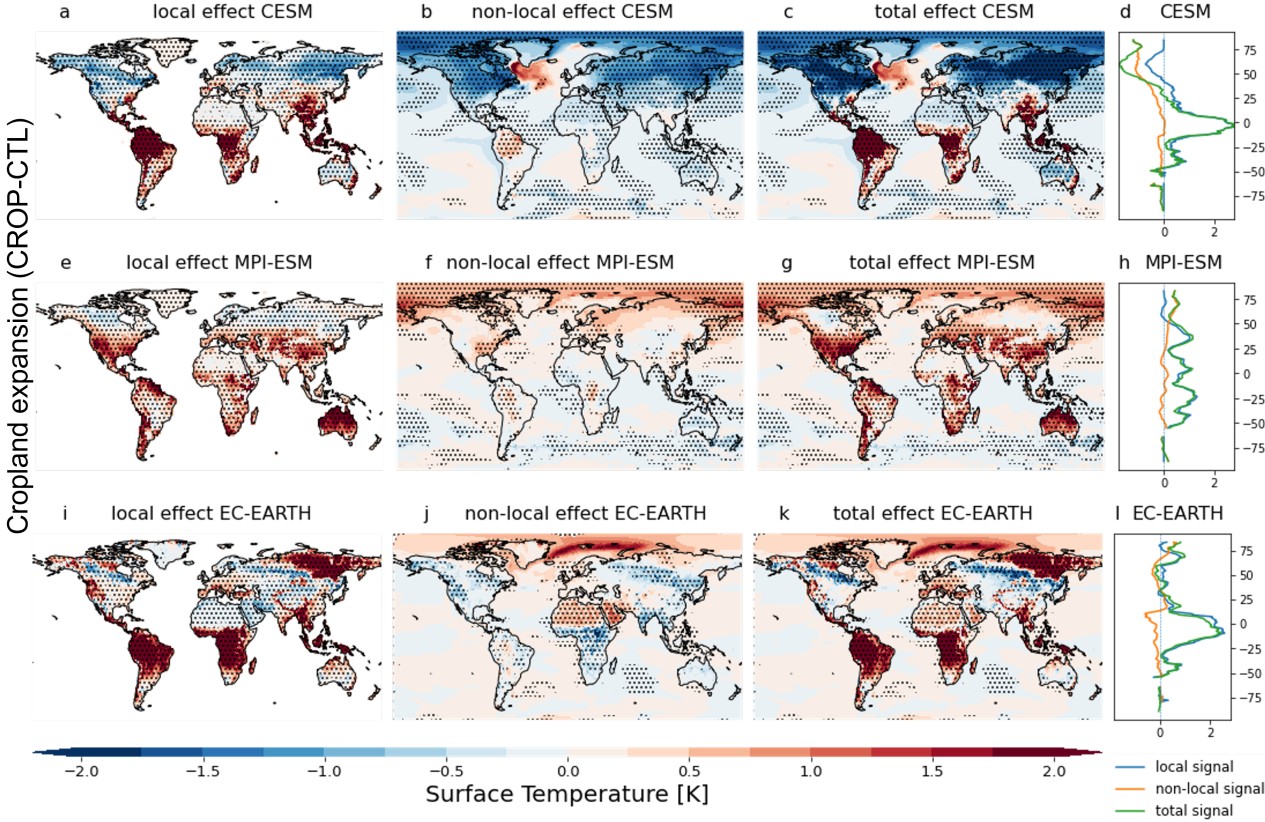

**Figure 5.** Annual mean surface temperature response to cropland expansion (CROP-CTL) of CESM (top row), MPI-ESM (middle row) and EC-EARTH (bottom row). For CESM: the local effect (a), the non-local effect (b) and the total effect (c), the global latitudinal average of the local (blue), non-local (yellow) and total (green) signals (d). (e-h): same as (a-d), but for MPI-ESM. (i-l): same as (a-d), but for EC-EARTH. The stippling on the maps shows grid cells where all 5 ensemble members agree on the sign of change.

are mixed, with a cooling over the (deforested) boreal forests and a strong warming over the permafrost covered areas (Siberia,
365 Northern Canada and Alaska). This NH boreal warming is most likely due to the shift in the EC-EARTH simulation from natural land to managed land, leading shorter duration of frozen soils throughout the year which causes a soil warming.

In CESM the local cooling is amplified by a strong non-local cooling over these regions. The non-local effect in MPI-ESM strongly differs from CESM. While CESM simulates a widespread cooling, MPI-ESM shows a weaker but clear warming over
370 the boreal regions, Europe and Eastern USA. The non-local effect in EC-EARTH is mixed with a warming over the Arctic regions and the Sahara, and a cooling in the mid-latitudes and tropics. In all ESMs the local signals dominates the total response in the tropics. The non-local effect also dominates over NH boreal latitudes in CESM and MPI-ESM while in EC-EARTH the



pattern differs regionally.

### 3.2.2 Afforestation

In the afforestation sensitivity experiment (FRST-CTL), the local response is similar to the response in the cropland expansion sensitivity experiment, but shows an opposite sign, as expected (Figure 6). A local cooling is simulated over the tropics for all ESMs and a local warming over the boreal latitudes for both MPI-ESM and CESM. The shift from cooling to warming occurs at a higher latitude in MPI-ESM and EC-EARTH compared to CESM. The lack of local boreal warming in EC-EARTH is probably related to the differences in experimental setup and the resulting low amounts of afforestation in this simulation (Figure 1f). The non-local effects due to afforestation result in warming for all ESMs, except over the North Atlantic in CESM. This indicates that the non-local effect is dominated by the albedo decrease, which originates from the strong snow masking effect of forest compared to open cropland. This is also indicated by the fact that the non-local warming dominates over the extratropics for all ESMs, in contrast to the local cooling which dominates over the tropics and parts of the subtropics (depending on the ESM).

In CESM, this albedo-induced warming causes a cooling blob in the North Atlantic (Figure 6b). A similar but opposite pattern is also apparent in the cropland expansion experiment with CESM (Figure 5b), but appears as a warming blob with lower magnitude. The same warming blob was also found in the LUMIP deforest-glob experiments by Boysen et al. (2020). A plausible explanation for this dynamic is the different latitudinal effect of the LCLMC option. With a high-latitude hemispheric warming and a slight cooling in low latitudes, the thermodynamic response of the Atlantic Meridional Overturning Circulation (AMOC) would indicate a weakening due to a decrease in the temperature gradient, similar to thermodynamic driven AMOC weakening due to arctic amplification under climate change scenarios (Schleussner et al., 2014). Inversely, global-scale cropland expansion causes non-local cooling except for a localised warming over the North Atlantic. It should be noted, however, that this strong North Atlantic response in CESM is not consistent throughout the entire simulation period despite its high magnitude. The global non-local warming pattern has large implications for future deployment of land-based mitigation strategies, especially for boreal afforestation. However, it should be noted that non-local signals are highly dependent on the spatial pattern as well as the extent of the prescribed land cover change (Winckler et al., 2019a)

### 3.2.3 Irrigation expansion

In the idealised irrigation expansion sensitivity experiment (IRR-CROP, i.e. irrigation expansion in a full cropland world), both MPI-ESM and CESM agree on the irrigation-induced reduction in local surface temperature, while irrigation expansion in EC-EARTH does not induce any local effects (Figure 7). The very limited local effects in EC-EARTH are caused by a lack of moisture exchange between IFS and LPJ-GUESS, whereby water added in LPJ-GUESS for irrigation does not affect the moisture fluxes in IFS. Hence, in EC-EARTH, irrigation affects crop growth and albedo but does not alter turbulent surface fluxes.



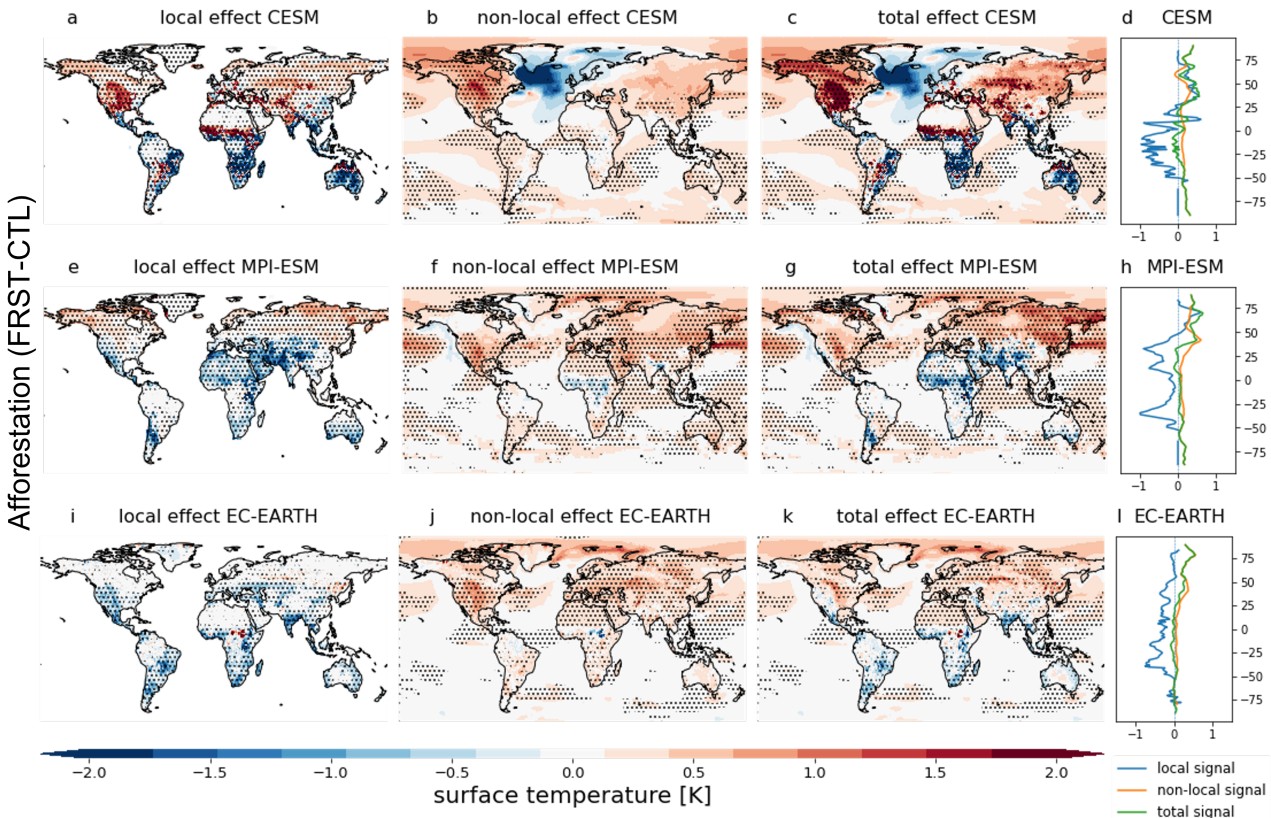

**Figure 6.** Same as Figure 5 but for afforestation (FRST-CTL).

In MPI-ESM and CESM, temperature decreases globally due to irrigation expansion, but there are substantial differences in the spatial patterns between the models. These differences partially stem from the large differences in irrigation amounts imposed in the different models (Figure 1i-k). EC-EARTH shows some non-local temperature effects but these are small in magnitude and the sign differs across different regions. In CESM, the total signal is dominated by the local response, with only a modest contribution of non-local effects. The non-local irrigation signal in MPI-ESM is generally stronger than the local signal and dominates the total response.

These results corroborate the findings of Thiery et al. (2017) and Chen and Dirmeyer (2019), who found that irrigation has a cooling potential due to an increased latent heat flux over irrigated areas. CESM simulates strong local cooling effects in the subtropics and tropics, while MPI-ESM shows the strongest local cooling in the NH mid-latitudes and less apparent local cooling in the tropics. In CESM, there is a non-local irrigation-induced cooling over the NH mid-latitudes where the local effects are generally small. This indicates that in these latitudes a non-local effect, plausibly due to an increase in cloud cover, dominates the effects of irrigation rather than surface processes like evaporative cooling, which dominate the local effects over





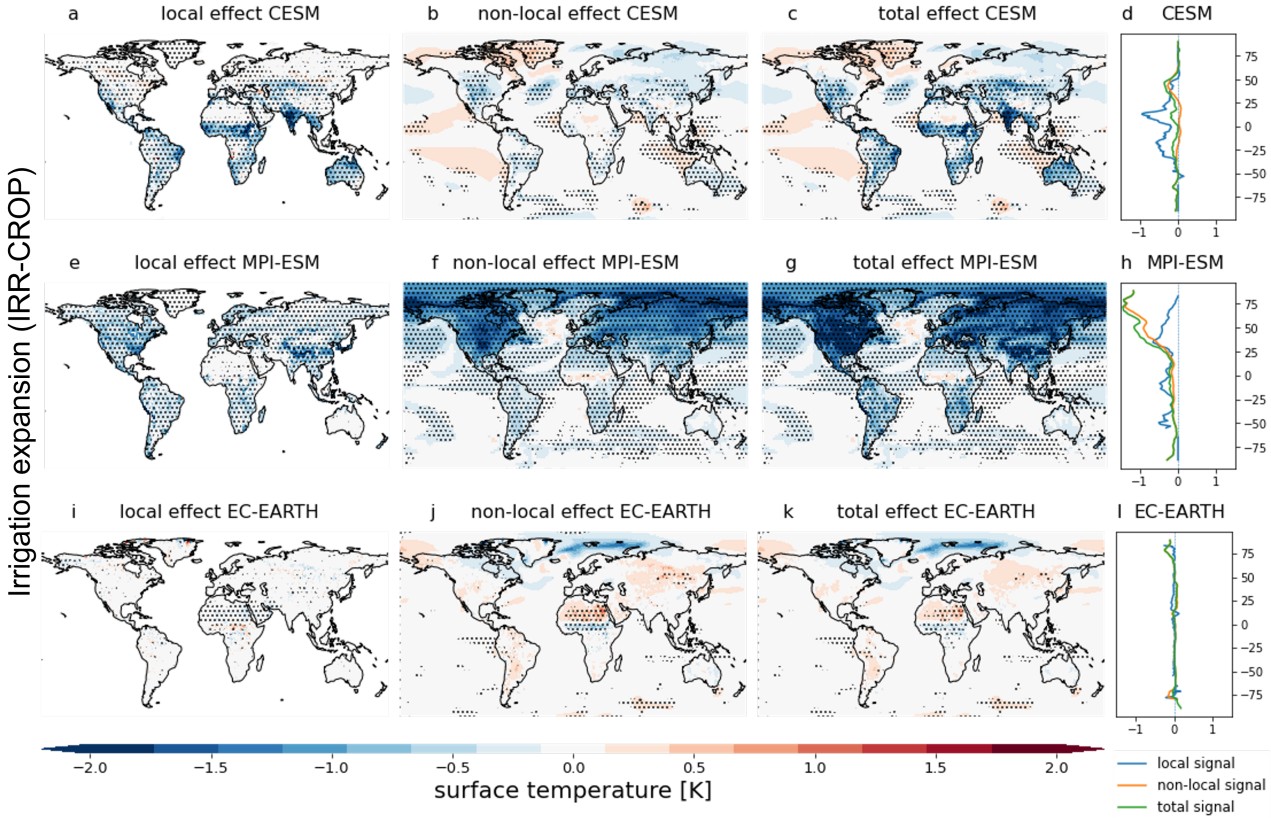

**Figure 7.** Same as Figure 5 but for irrigation expansion in a cropland world (IRR-CROP).

the tropics. For MPI-ESM, a strong increase in cloud cover appears to cause the strong non-local cooling (Figure E15).

420

### 3.2.4 Wood harvest expansion

The effect of wood harvesting (HARV-FRST) appears to be very small (Figure 8). There is generally no local effect and the non-local signal is overall weak and inconsistent in sign across the CESM simulation. The simulated non-local signals may well stem from internal climate variability rather than an actual response to land management change. These results imply that

425 the biogeophysical effects of wood harvesting, as simulated here, are too weak to have a significant imprint in global and local climate conditions at the grid scale in the represented ESMs. This does not imply that the biogeophysical effects cannot play a role locally, but simply suggests that these effects are not strong enough to be discerned at the currently used grid scale level and with the process-detail of current ESMs. An analysis comparing the simulation results at the tile level (within a grid cell) would provide an alternative approach to analyse possible local effects due to wood harvesting.





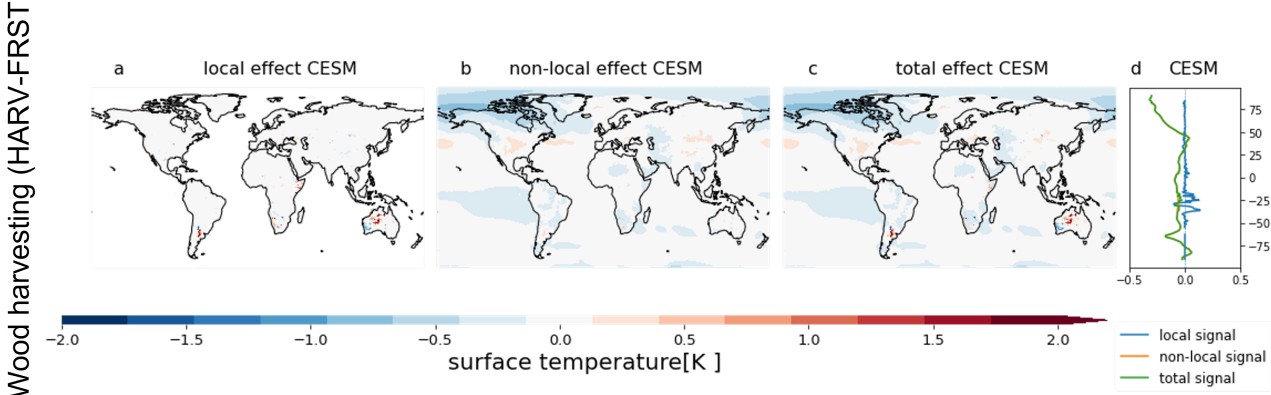

**Figure 8.** Same as Figure 5 but for wood harvest expansion (HARV-FRST). Only results for CESM are shown as MPI-ESM does not simulate biogeophysical effects of wood harvesting and EC-EARTH did not conduct these simulations.

**Table 4.** Summary of local and non-local effects due to the different LCLMC. Each cell indicates where the changes in surface temperature response are consistent in sign.

| LCLMC | Local effects | Non-local effects | Total effects |
|---|---|---|---|
| cropland expansion | tropical warming | none | tropical warming |
| afforestation | tropical cooling | global warming | warming across boreal latitudes and cooling over tropics |
| irrigation expansion | regional cooling | regional cooling | regional cooling |

## 3.3 Energy balance decomposition of the surface temperature changes

### 3.3.1 Cropland expansion

Using Equation 2, the different factors contributing to the response in surface temperature are assessed when aggregated zonally (Figure 9) and seasonally (Figure 10). In the case of cropland expansion, the warming in the tropics for all ESMs is mostly caused by a strong decrease in latent heat flux, possibly as a consequence of a decreased evaporation capacity (Figure 9a and b). The simulated decrease in sensible heat flux in MPI-ESM reduces the heat transport away from the surface, therefore amplifying the warming, while in CESM an increase in sensible heat contributes to a cooling. In MPI-ESM the tropical warming is slightly offset by an albedo increase. In all ESMs, local changes in shortwave and longwave radiation increase the warming signal, however, in EC-EARTH the contribution from enhanced incoming longwave radiation is especially strong, which could indicate that atmospheric properties such as high cloud cover or atmospheric moisture have a strong influence on surface temperature in this model. In CESM over boreal latitudes, the increase in albedo dominates the surface temperature response




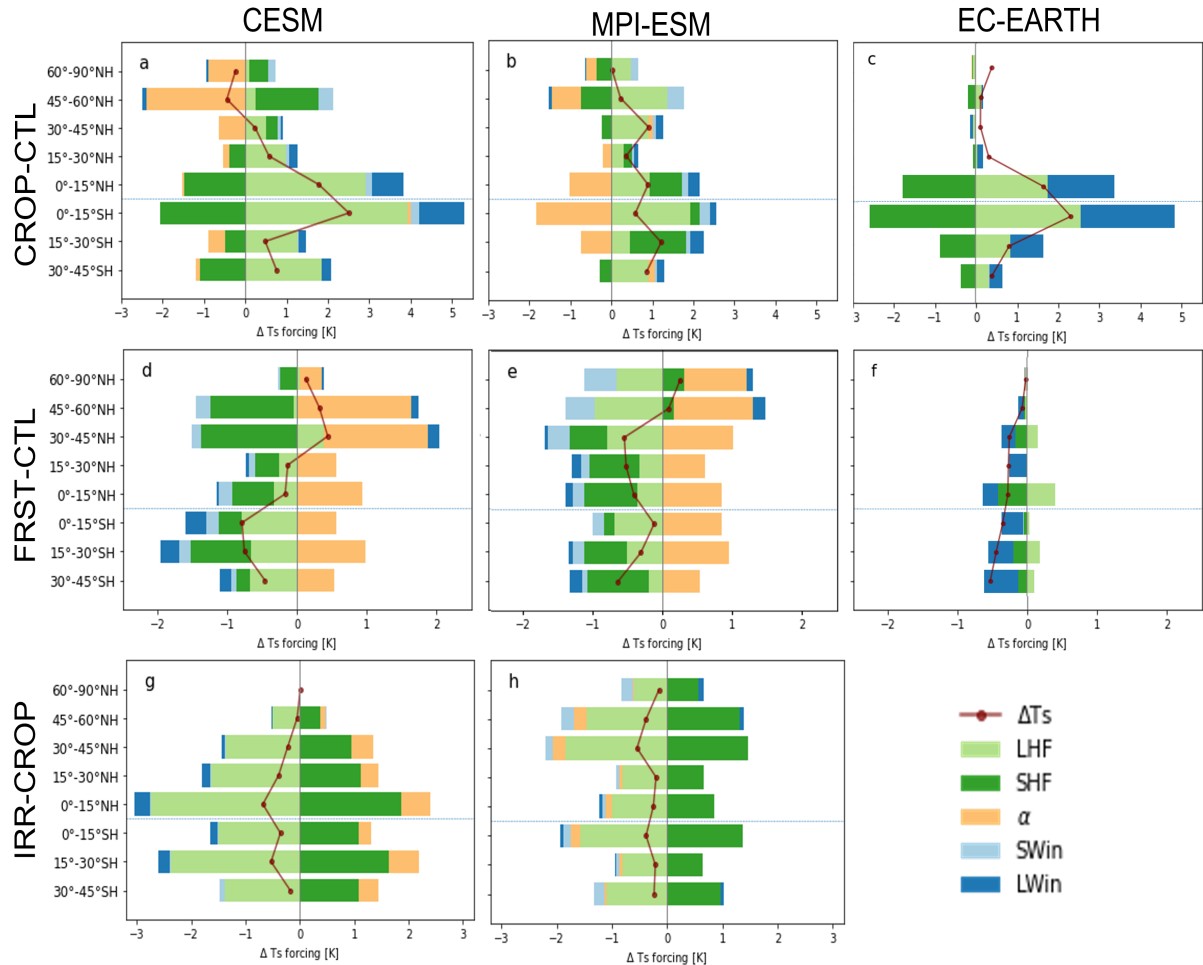

**Figure 9.** The energy balance decomposition of the local surface temperature for the different latitudinal bands. The response to cropland expansion (CROP-CTL) for CESM (a), MPI-ESM (b), and EC-EARTH (c), the response to afforestation (FRST-CTL) for CESM (d), MPI-ESM (e), and EC-EARTH (f) and the response to irrigation expansion (IRR-CROP) for CESM (g) and MPI-ESM (h). EC-EARTH is not shown for irrigation expansion as the local effects are too small for any meaningful analysis.

causing a local cooling which is partly offset by a warming induced by a decrease in sensible heat flux. In MPI-ESM, this boreal albedo effect is much weaker causing no clear local cooling.

In EC-EARTH, the energy balance components do not explain the simulated warming over boreal latitudes, which is most likely related to the fact that EC-EARTH uses the temperature of the first whole soil layer as surface temperature. As a consequence, other processes that are not related to the surface energy balance (e.g. permafrost thawing) also affect the surface temperature in this model. Finally, contrasting to the other models, the albedo in EC-EARTH does not influence the local




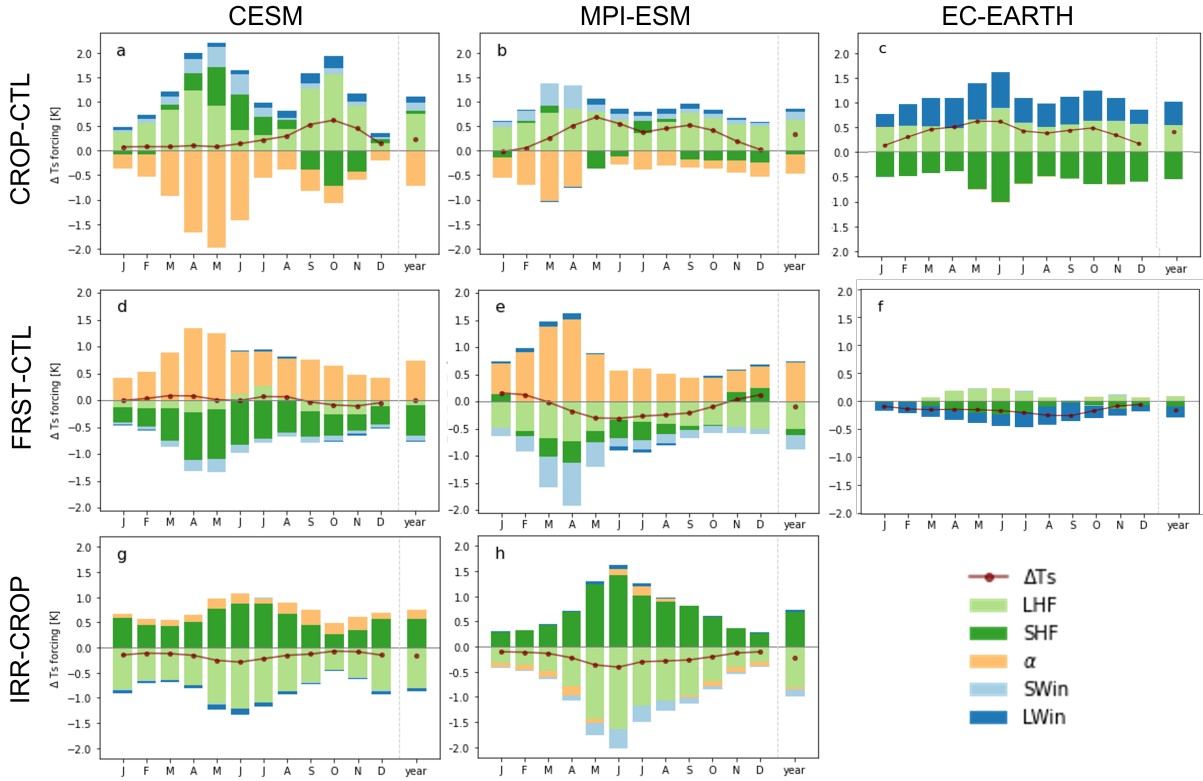

**Figure 10.** Global average seasonal cycle of energy balance decomposition of local surface temperature. The response to cropland expansion (CROP-CTL) for CESM (a), MPI-ESM (b), and EC-EARTH (c), the response to afforestation (FRST-CTL) for CESM (d), MPI-ESM (e), and EC-EARTH (f) and the response to irrigation expansion (IRR-CROP) for CESM (g) and MPI-ESM (h).

surface temperature changes, as there is no change in local albedo (see Figure E2).

The cooling effect of albedo due to cropland expansion has a pronounced seasonal response in both MPI-ESM and CESM (Figure 10a and b). It is most outspoken during NH spring as a consequence of the reduced snow masking effect. In both MPI-ESM and CESM, the latent heat flux has a strong contribution throughout the year. It shows a seasonality which is most pronounced in CESM, peaking in early spring and fall. The sensible heat flux has a warming effect in CESM throughout most of the year except during the NH fall when it shows a cooling effect. In EC-EARTH the sign of all changes is constant through-

out the year. There is a slight seasonal effect for the magnitude of the turbulent heat fluxes and longwave incoming radiation being largest in NH summer and lowest in NH winter. Overall, all ESMs simulate a global surface warming of about 0.3 K due to the local effect of cropland expansion over the year and show a minimal warming in the NH winter.





### 3.3.2 Afforestation

In the case of afforestation, all models show a reduction of the surface temperature in the SH and tropics (Figure 9d,e,f). In MPI-ESM and CESM, this is caused by the cooling effect of increasing turbulent heat fluxes, which is partly counteracted by a warming effect due to an albedo decrease. This albedo effect becomes dominant when moving northward and causes a local warming in CESM starting from the mid-latitudes and in MPI-ESM starting from the boreal latitudes. In EC-EARTH, the cooling is caused by changes in sensible heat flux and incoming longwave radiation, but is counteracted by a decrease in latent

heat flux. At boreal latitudes, the albedo-induced warming is partly counteracted by an increase in sensible heat flux in CESM, and by an increase in latent heat flux and a decrease in incoming shortwave radiation in MPI-ESM. The decrease in incoming shortwave radiation might be caused by an afforestation-induced local increase in cloud cover (as shown in Figure E10). This would be in line with the theoretical understanding that an increase in latent heat flux causes an increase in low cumuliform clouds (Ban-Weiss et al., 2011). Recent observational results show an afforestation induced cooling effect related to increased

cloud cover (Teuling et al., 2017; Duveiller et al., 2021). However, neither CESM nor EC-EARTH represent this increase in cloud cover, with CESM even showing a slight decrease in cloudiness over boreal latitudes (Figure E10).

The albedo-induced effect of afforestation has a clear seasonal peak during NH spring for both MPI-ESM and CESM (Figure 10c and d). The turbulent heat fluxes seem to follow a similar seasonality. This indicates that extra-tropical afforestation is

dominating the global climate response for these models due to a strong albedo response largely counteracted by the changes in turbulent heat fluxes. In EC-EARTH, a similar seasonal pattern is visible with larger fluxes in NH summer and smaller fluxes in NH winter as was also the case for cropland expansion. Overall, all models show limited local effects due to afforestation, being quasi 0 K in CESM, -0.15 K in MPI-ESM and 0.2 K in EC-EARTH.

### 3.3.3 Irrigation expansion

For irrigation, only results for MPI-ESM and CESM are shown as the local surface temperature changes in EC-EARTH are too small for a meaningful decomposition in energy balance components. Both MPI-ESM and CESM show a very different geographic pattern for the irrigation flux (Figure 1). However, the models appear to be largely consistent when it comes to the identification of the underlying processes causing the change in surface temperature (Figure 9e and f). The increase in latent

heat flux dominates the response. This causes a strong cooling which is counteracted by a strong (but weaker) warming effect caused by the decreased sensible heat flux. Surface albedo increases slightly in CESM as wet soils are darker. This change contributes to a rise in surface temperature. MPI-ESM, in contrast, shows a slight decrease in albedo contributing to a lowering of surface temperature. We hypothesise that this albedo decrease in MPI-ESM is a consequence of irrigation causing greener, hence brighter, crops. Longwave and shortwave radiation both give a cooling contribution due to a local increase in cloudiness

(Figure E15).



The seasonal pattern of irrigation is dominated by the application of irrigation during the dry season (Figure 10e and f). As most land is located in the NH, we find the strongest local cooling during NH spring and summer. This seasonal pattern is stronger in MPI-ESM as irrigated croplands extend more northward than in CESM (Figure 1g,h). Globally both models predict
a slight global cooling effect of around 0.2 K.

## 4   Discussion

### 4.1   Robust patterns in the local response to LCLMC across ESMs

Our results show clear consistencies across CESM, MPI-ESM and EC-EARTH. All three ESMs are able to simulate a response of average annual surface temperature to full deforestation consistent with observational evidence. There remain some
clear biases when comparing the ESMs to observations such as a strong albedo response in CESM in the mid-latitudes and a strong (soil-related) warming response in the high latitudes in EC-EARTH. However, general observed patterns such as local cooling over boreal forests and local warming over tropical forests are well captured by the ESMs. The consistency in surface temperature response across ESMs and observations is in stark contrast to the large spread in signals of the turbulent heat fluxes and albedo, which have been highlighted as some of the main driving processes of local temperature change (Davin and
de Noblet-Ducoudre, 2010; Winckler et al., 2019c). The energy balance decomposition for the cropland expansion confirms these model biases, which moreover differ across ESMs. For afforestation and cropland expansion, all ESMs show that the tropical response is mainly caused by a change in turbulent heat fluxes. However, they disagree on how these changes occur. All three ESMs show that local latent heat flux changes determine the surface temperature response in the tropics. However, the role of local sensible heat flux changes differs across ESMs, showing a cooling effect in CESM and EC-EARTH in contrast
to MPI-ESM where it has a warming effect. Over boreal latitudes, the albedo dominates the local effect for both cropland expansion and afforestation in CESM and for afforestation in MPI-ESM. EC-EARTH shows that permafrost thawing (unrelated to land cover change) is causing the simulated warming in the cropland expansion experiment. For irrigation expansion, MPI-ESM and CESM consistently show that the increase in latent heat dominates the surface temperature response, causing a local cooling. In EC-EARTH, the moisture fluxes to the atmosphere caused by irrigation are not modelled, hence there is no
clear effect.

Although we have harmonised the land cover and management representation across the different models, strong differences remain, most notably in the implementation of irrigation expansion and afforestation (Figure 1). This implies that the comparison of the different simulations across ESMs is not perfect and inconsistencies can be caused by disparity in model structure
and by spatial differences and differences in extent of the applied LCLMC. As for afforestation, the differences found here were mainly caused by the technical difficulty of implementing this in the dynamic vegetation model LPJ-GUESS used in EC-EARTH. However, the differences regarding land management are a direct consequence of these implementations being fairly recent in the various ESMs. There is no consistency in the implementation approach for land managements such as irrigation expansion across ESMs, as was also the case in the early land cover change inter-comparison projects (De Noblet-





Ducoudré et al., 2012). Over the last decade several improvements were made regarding land cover change to make the ESMs more consistent. For example, using common datasets (Hurtt et al., 2020) and common simulation protocols like the LUMIP experiments under CMIP6 (Lawrence et al., 2016). The same issues that ESMs faced before for land cover change are now apparent for land management change as well. As more ESMs are implementing land management change (Blyth et al., 2021), it is crucial that common datasets and simulation protocols are set up in order to ensure comparability across the various ESMs.


However, despite these limitations our results show that there remain similarities in the LCLMC response in the different ESMs, most notably regarding the local effects. A consensus is emerging regarding the local effects of deforestation/afforestation with a clear cooling/warming at boreal latitudes and a warming/cooling in the tropics, as is in line with observational evidence. The cooling potential of irrigation (both local and non-local) is confirmed by both MPI-ESM and

CESM. However, more research is needed to understand the full implications of these biogeophysical effects. The cooling effects induced by irrigation might be offset by the increased humidity and overall induce an increase in heat stress (Mishra et al., 2020). The effects on warm and cold extremes remain to be investigated as well, but lie beyond the scope of the current study.

Our results highlight the importance of including possible local biogeophysical effects in future land-use and land-management policies. The current policies underpinning large-scale climate mitigation plans such as the European Green Deal are set up to only take into account the biogeochemical effects of LCLMC strategies such as afforestation. The European Green Deal plans (European Commission, 2020) rely heavily on afforestation as a possible negative emission technology to enhance the land sink by planning to plant up to 3 billion trees within the EU. However, beyond the positive consequences of afforestation

on carbon storage, its biogeophysical effects should also be considered in order to plan for (or avoid) side-effects for regional temperature induced by local processes (as shown in Figure 6a,e,i). The local biogeophysical effects imply some positive side-effects over specific regions, such as the tropics and mid-latitudes, especially during the summer season; however they could also imply some negative side-effects over the boreal latitudes and part of the mid-latitudes during the winter season. These findings are in line with Windisch et al. (2021) who highlight the existence of various trade-offs between local biogeophysical

effects and biogeochemical effects depending on the season and region. These results further strengthen the need for the inclusion of local biogeophysical effects next to biogeochemical effects in order to have an accurate idea of the mitigation potential of forests in LCLMC policies.

### 4.2    Inconsistent non-local effects across ESMs due to idealised cropland expansion

The global non-local cooling in CESM as shown in Figure 5 is consistent with the findings of a previous global deforestation

simulation using the checkerboard approach performed by Winckler et al. (2019a) with MPI-ESM. However, these results strongly contrast with the non-local response found in MPI-ESM here. Some methodological differences should be noted here: Winckler et al. (2019a) performed a fully idealised deforestation experiment, which is more akin to CROP-FRST comparison in this study than the results of CROP-CTL shown here. It should be noted that for full deforestation (i.e. CROP-FRST) all ESMs





(including MPI-ESM) predict a non-local cooling (Figure C3), which is consistent with Winckler et al. (2019a). The effect
of a cropland expansion (CROP-CTL) in MPI-ESM, which starts from present-day forest extent, results in a clear non-local
boreal warming. Two possible mechanisms could explain this counter-intuitive discrepancy between the non-local response of
MPI-ESM and CESM in CROP-CTL, in contrast to their consistent results for CROP-FRST: (i) MPI-ESM shows a weaker
albedo effect when compared to CESM (Figure 5c), additionally (ii), the MPI-ESM model shows a strong decrease in annual
boreal cloud cover (see Figure E5), which is especially strong in boreal summer (not shown) and could cause an additional
warming, possibly offsetting any non-local cooling caused by changes in albedo.

In summary we can state that the non-local effects due to full deforestation presented here are in line with literature (Winckler et al., 2019a). However, the non-local effects display a larger uncertainty when it comes to the non-local effects of cropland
expansion from present day conditions (i.e. CROP-CTL as presented here). It should be noted that due to the strong albedo
bias in CESM over NH mid-latitudes (see Figure 4d) and the crucial role of albedo in determining the non-local effects, it is
probable that the strong non-local cooling shown over CESM is an overestimation.

### 4.3 Non-local biogeophysical response due to land-based mitigation and adaptation

Non-local biogeophysical effects can regionally dominate over local biogeophysical effects. The distinct non-local warming
found for afforestation is consistent with the inverse outcome obtained from global deforestation experiments in literature
(Winckler et al., 2019a; Davin and de Noblet-Ducoudre, 2010) and is robust across the different ESMs considered here (Figure 6b,f,j). However, the strong divergence in outcome from the cropland expansion experiments do show that the albedo effect
does not completely control the non-local surface temperature responses. A variety of atmospheric processes affecting the atmospheric moisture balance and large scale atmospheric dynamics need to be assessed in order to better understand the relevant
processes. In CESM a large scale land cover change even appears to affect global ocean circulation, as was illustrated by the
strong AMOC response within this model. It should be noted that this is not a single model feature, as similar AMOC anomalies were visible for 2 other ESMs in the LUMIP deforestation simulations (Boysen et al., 2020). More research is needed to
fully understand the processes that cause the non-local biogeophysical effects related to large scale land cover change shown
here.

Irrigation clearly decreases temperature in both CESM and MPI-ESM, constituting another demonstration that deploying
irrigation could entail side-benefits for local temperature reduction especially over agricultural land (Thiery et al., 2017, 2020;
Hirsch et al., 2017). These results even suggest that achieving climate benefits could become an objective of irrigation deployment, making it potentially a deliberate adaptation strategy if constraints to its implementation (related for example to
water availability or socio-economic enabling conditions) can be overcome. However, it remains unclear whether the irrigation
induced cooling is predominantly local (induced by turbulent heat fluxes) or non-local (induced by cloud effects), and what the
combined effect is of irrigation-induced changes in temperature and humidity patterns on heat stress. Nevertheless, these results





help assess the future climate consequences of irrigation expansion. Irrigation has been projected to increase in the future as a means to increase agricultural productivity (van Maanen et al., 2022; Rosa et al., 2020) but it may also aggravate future water
stress (Haddeland et al., 2014). It should be noted that irrigation is implemented in a highly idealised way in these simulations, with 2 out of 3 ESMs not being constrained by water limitations. These water limitations should be assessed before irrigation expansion can be considered as a viable adaptation option in any region.

Overall, our results show that future land-based mitigation strategies will need to consider the non-local biogeophysical con-
sequences of LCLMC patterns, as large scale afforestation is a key strategy in intensive land-based mitigation scenarios (Smith et al., 2015; Humpenöder et al., 2014), especially in those compatible with a 1.5 K world (Roe et al., 2019). In particular, the robust non-local biogeophysical warming from global afforestation presented in this study indicates that future land-based mitigation strategies would lead to an even more extensive unintended warming than the local biogeophysical warming that has been widely reported for boreal regions and the mid-latitudes in winter. More research is needed to the bridge knowledge
gaps regarding which regions would be mostly responsible for this non-local warming if afforested and what would be the magnitude of this warming in realistic afforestation scenarios.

## 4.4   Limitations and outlook

The idealised simulations performed in this study give an overview of the potential biogeophysical effects from LCLMC. We
were able to separate local and non-local effects due to the application of a checkerboard like LCLMC perturbation to our idealised land cover maps (Figure 1). The local effects are only caused by changes occurring within the grid cell. Hence, they represent the most extreme possible outcome of the application of a certain LCLMC within that single grid cell, without accounting for other LCLMC around the globe. In contrast, the non-local signals are a compound response caused by the LCLMC around the globe. These represent an underestimate in magnitude of the non-local effects in a simulation of global
LCLMC, as due to the checkerboard pattern, non-local effects are the consequence of LCLMC applied to only half of the grid cells around the globe. As the non-local effects, by design, also capture all internal climate variability they are more uncertain than the local effects presented here. To limit the uncertainty related to climate variability as much as possible, the simulations could be repeated within an ensemble setup. However, such setup would require substantial additional computation and storage resources.

Furthermore it should be noted that the application of the checkerboard approach has some methodological implications, as the resulting local and non-local signals intrinsically contain an interpolation error. Although we tried to minimise this error by using a checkerboard pattern of 1 out of 2 grid cells, this error can still reach up to 0.3 K based on previous simulations with MPI-ESM (Winckler et al., 2017). Moreover, the approach has limitations due to the size of a grid cell in the different ESMs.
The land cover change needed to get a local effect as presented here remains highly unrealistic (around 100 km). As ESMs are becoming computationally more efficient and their resolution gets increased, the validity of this assumption could be tested



using higher resolution ESMs.

Some biases exist within the evaluation approach as the modelled surface temperature does not exactly match the radia-
tive surface temperature measured in the observational estimates. For instance, the satellite measurements have an inherent
sampling bias as they only measure during cloud free conditions. Also, the different observational estimates have different
and often non-overlapping spatial coverage. Nevertheless, these observational studies using a diversity of approaches show a
large consistency among themselves and thus can act as a benchmark for the representation of land cover change within ESMs
(Winckler et al., 2019b, a).


The results shown within this paper highlight some clear consistencies across the ESMs, however, often the ESMs tend to
show differences as well. For example, more work needed to improve the representation of irrigation, especially for EC-EARTH
and MPI-ESM. As MPI-ESM suffers from unrealistic irrigation amounts, especially in the boreal regions while underestimat-
ing the potential irrigation in the subtropics such as India. Furthermore, EC-EARTH is currently not a viable model for a study
of the biogeophysical effects of irrigation, as water fluxes from land are not communicated to the atmosphere. This limitation
is worth addressing as the implementation of irrigation in ESMs has been shown to make them more realistic over regions of
intense irrigation (Al-Yaari et al., 2022). Regarding land cover change, all ESMs still struggle to replicate observed patterns
in energy fluxes (Figure 4). CESM has a strong overestimation of the albedo in the intermediate latitudes (30°N-50°N) with
clear temperature biases over these regions, an issue which could be considered in future development of this ESM. For EC-
EARTH, even though it has a highly advanced land model (LPJ-GUESS), the interface with the atmosphere is handled by a
more simple submodel (HTESSEL) within the atmosphere model IFS. This causes some clear biases such as the unrealistic
partition of albedo as a non-local feature in EC-EARTH (Figure 4c). Addressing these biases could be a useful strategy when
further developing this ESM to make land cover induced climate effects more realistic.

The simulations presented here are unique as they combine a multi-model approach with a direct separation of local and
non-local effects. Further analyses could investigate the effects of LCLMC beyond the seasonal and mean changes in surface
properties, heat fluxes, and temperature. These simulations allow to analyse both the transient response of LCLMC induced
biogeochemical effects, and the socioeconomic impact from their biogeophysical effects. The non-local effects presented here
can further be analysed to gain a better understanding of the circulation changes induced by the LCLMC. A moisture tracking
analysis could be performed to investigate the effects on global precipitation patterns, as previous studies showed that Amazo-
nian deforestation could induce a drying of the region (Lejeune et al., 2015). The local effects diagnosed from these extreme
sensitivity experiments could also be used as training data for less computationally expensive statistical models to emulate
biogeophysical effects arising from less extreme and more realistic LCLMC scenarios. Overall, we hope that the results of
the simulations presented here can help increase the present understanding of LCLMC and build towards a framework that
facilitates the inclusion of biogeophysical effects of LCLMC in future policy frameworks.



# 5 Conclusions

In this study, we showed the first results of a new slate of fully-coupled ESM simulations within a multi-model framework targeted at analysing the effects of land cover and land management change (LCLMC). We simulate the global biogeophysical response to (i) cropland expansion (ii) afforestation, (iii) irrigation expansion and (iv) wood harvesting, using the Community Earth System Model (CESM), the Max Planck Institute Earth System Model (MPI-ESM) and the European Consortium Earth System Model (EC-EARTH). We apply the checkerboard approach of Winckler et al. (2017) to disentangle the local and non-local biogeophysiscal effects.

A model evaluation is performed for a global deforestation scenario using the local effects derived from the ESM simulations and several observational estimates. All ESMs agree well with the observed annual mean surface temperature change. CESM, however, overestimates the albedo in boreal and mid-latitudes, and persistently locates the transition from local warming to local cooling more south compared to the observations. A soil-induced effect in EC-EARTH causes a warming in boreal latitudes. MPI-ESM and EC-EARTH show strong differences in the representation of the turbulent heat fluxes despite their overall agreement with observed surface temperature changes.

The biogeophysical effects of idealised LCLMC are shown to be important and non-negligible to understand the overall climate impact of LCLMC. Deforestation causes a local warming in the tropics and a cooling over boreal latitudes for all ESMs. For afforestation, a clear tropical cooling is consistent across ESMs. The non-local effects carry more uncertainty which may be due to a wider variety of mechanisms at play and due to the strong natural variability intrinsic to atmospheric processes. However, this would require further investigation to be confirmed. All ESMs show a strong non-local warming as a consequence of large-scale afforestation. Irrigation expansion cools the climate both through local and non-local effects, although the contribution of local and non-local effects to this cooling is inconsistent across ESMs. Finally, the effect of extensive wood harvesting is shown to be too small to have a clear imprint on the grid-scale climate.

The driving processes underlying the local surface temperature effects were analysed using an energy balance decomposition technique. The local surface temperature effects of land-cover change (both cropland expansion and afforestation) are dominated by the response in turbulent heat fluxes in the tropics. In the case of afforestation, the albedo is the dominant factor in boreal latitudes for MPI-ESM and CESM. This is also the case for the local effects in the cropland expansion experiment for CESM, in contrast to the MPI-ESM where turbulent fluxes dominate in the boreal latitudes. In EC-EARTH, the boreal surface temperature change could not be explained by the energy balance decomposition, as the boreal warming is caused by processes that are not included in the simplified version of the surface energy balance, such as permafrost thawing. Moreover, the strong influence of incoming longwave radiation indicates that atmospheric properties (such as cloud cover and moisture content) are strongly related to local surface temperature changes. Both CESM and MPI-ESM agree that the main local surface temperature response due to irrigation is driven by a strong increase in latent heat flux which is only partly counteracted by a decrease in



sensible heat flux.

Overall, our results confirm that the biogeophysical effects of LCLMC are an important factor to consider in future land planning strategies, especially as they reveal the robust importance of non-local climate responses in the context of mitigation potential of land cover change. In the case of large-scale afforestation specifically, the non-local response could lead to global-scale unintended warming, in particular over the boreal and mid-latitude regions.



*Code and data availability.* CESM is an open source model which can be freely downloaded here (https://www.cesm.ucar.edu/models/cesm2/release_download.html). The scripts used for the signal separation of the 3 ESMs, the evaluation and the energy balance decomposition can be found on the github page of the hydrology department of VUB (https://github.com/VUB-HYDR/2022_De-Hertog_etal_ESD).

705 The simulation data used in this paper will be made available through the dkrz, for those interested in using these data please contact the authors.

## Appendix A:  Differences in forest fractions in CTL land cover maps

In Figure A1 the fraction of deciduous, evergreen and total forest cover are shown for the 3 ESMs. This is to illustrate the differences in the CTL land cover maps which stem from a different definition of the natural PFTs in each ESM. Although all

710 ESMs are based on the LUH2 dataset we can still see that there are clear differences in the types of forest modelled (evergreen or deciduous) but also in the total amount of forest.

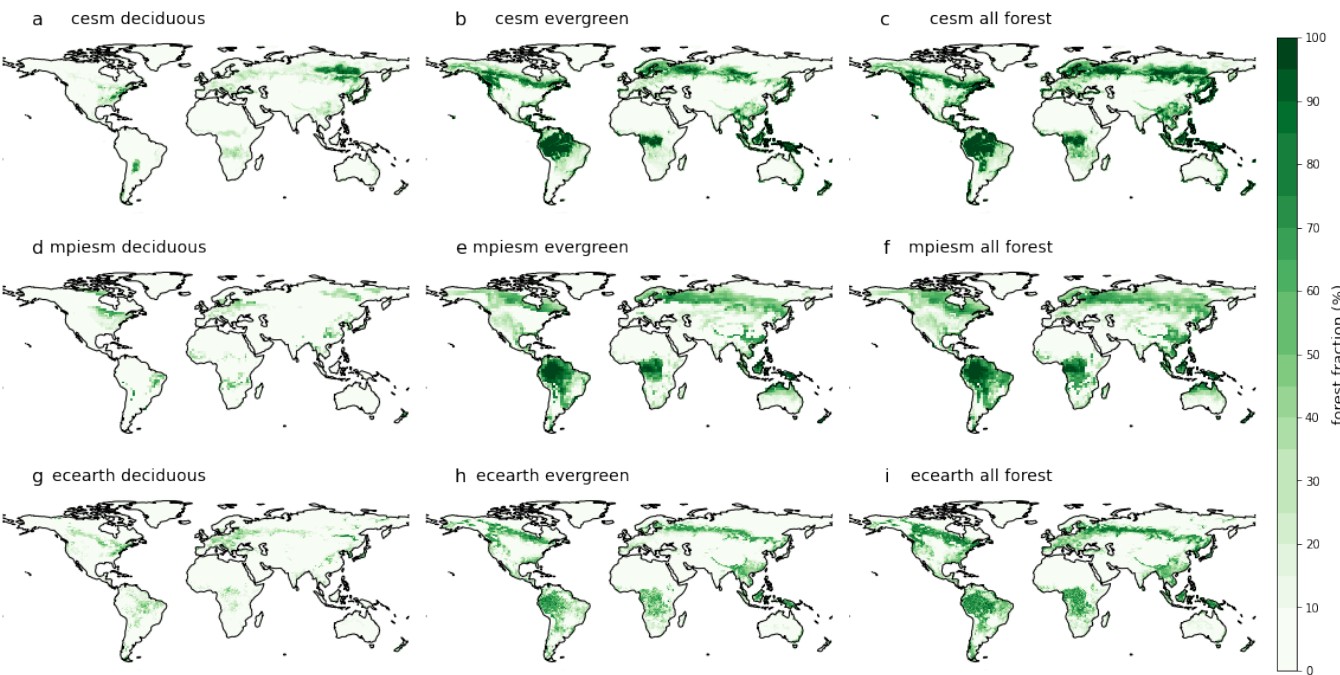

**Figure A1.** Total amount of forest (%) is shown for the 2015 CTL map for each ESM displaying different forest types. The amount of deciduous forest for CESM is shown in panel (a), the amount of evergreen forest in panel (b) and total amount of forest in panel (c). For MPI-ESM the amount of deciduous forest is shown in panel (d), evergreen forest in panel (e) and total amount of forest in panel (f). For EC-EARTH the amount of deciduous forest is shown in panel (g), evergreen forest in panel (h) and total amount of forest in panel (i).



## Appendix B: Irrigation implementation in the different ESMs

MPI-ESM:

- soil moisture of first (0 - 0.065 m) and second (0.065 - 0.319 m) soil layer (out of 5) is filled up each time step (20-30 minutes) to field capacity if field capacity was not reached and if enough irrigation water is available in storage.

- irrigation water is stored each time step when the reservoir drops below 0.2 m and filled up with all available water from (surface) runoff and drainage during that time step.

CESM:

-Irrigation is applied daily at the first timestep after 6AM only when the soil moisture over all soil layers containing roots falls below a defined target soil moisture which is defined in order to match present day irrigation. If soil moisture falls below the target soil moisture it is replenished until at the target soil moisture level.

-The water needed for applying irrigation is taken from river water storage, however when this is inadequate to meet water demand it can also be subtracted from the ocean model, therefore no real water availability limit is applied within CLM.

-Irrigation is only applied when the crop leaf area >0, i.e. this means that crops are only irrigated when they are in there vegetation state (during the growing season).

EC-EARTH:

-In LPJ-GUESS the amount of irrigation is the deficit a crop plant is experiencing. So if a crop needs an additional amount of water, it is added to the top of the soil column.

-The water comes from nowhere (i.e. unlimited water source).

- The water flux is not communicated to IFS, i.e. irrigation does not affect the surface water fluxes within the atmosphere. The only effect is that an irrigated crop would have a higher leaf area index and cover fraction compared to a non-irrigated crop of the same type.

## Appendix C: Surface temperature in observational datasets

The comparison of the ESM data and the different observational datasets has some inconsistencies as was already described before by Winckler et al. (2019a). From Figure C1 it is apparent that the different datasets do not have the same spatial coverage. Besides this the calculation of the temperature signal differs across studies. In Alkama and Cescatti (2016) the observed signal is extracted by looking at changes over time in contrast to the other studies where this was extracted by comparing nearby locations during the same timestep. Also different conversion types are considered, in Li et al. (2015) and Duveiller et al. (2020) a generic forest to open land (both crop and grassland) is considered while in Bright et al. (2017) only a forest to grass conversion is considered. In Alkama and Cescatti (2016) apart from forest clearing to grass and crop also windfall events and fires where included in the analysis. Each dataset also covers different time periods although all datasets only include data



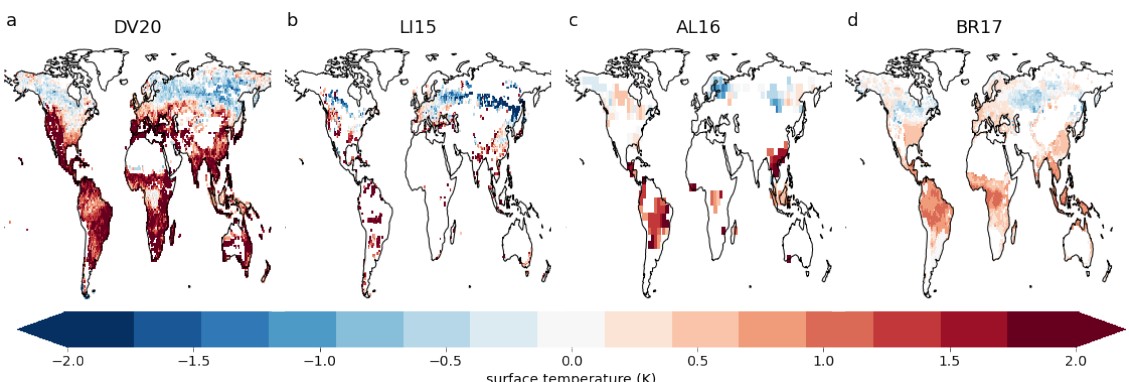

**Figure C1.** Annual mean surface temperature is shown for the used observational datasets. The data from Duveiller et al. (2020) is shown in panel a, from Li et al. (2015) is shown in panel b, from Alkama and Cescatti (2016) is shown in panel c and from Bright et al. (2017) is shown in panel d.

after the year 2000 (hence representing present day conditions) and the total duration each estimate is based on are similar. All studies provide an estimate of the response of surface temperature to a full deforestation except Alkama and Cescatti (2016) where actual deforestation was considered and which had to be converted to a full deforestation signal by weighting with the deforestation fraction, in order to get robust results only grid cells where selected where more than 1% of actual deforestation had occurred over the analysis period considered. For Bright et al. (2017) only data was provided for conversions from specific

forest species, to allow for a consistent comparison to the ESMs these values had to be weighted using the weights of each forest PFT within the specific ESMs. Therefore, an estimate of the Bright et al. (2017) data was created representing the different ESMs there PFT distributions, however, these differed only slightly so an average was taken over all estimates to be compared across all ESMs.

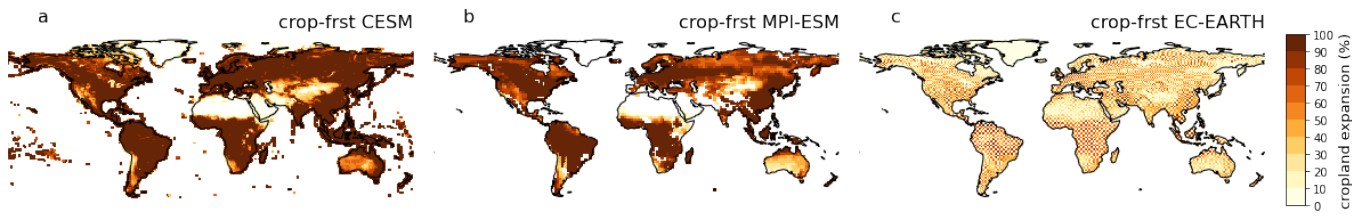

**Figure C2.** Total amount of deforestation (%) is shown for the CROP-FRST signal separated data for CESM in panel (a), MPI-ESM in panel (b) and EC-EARTH (c). Note that the land cover maps are not interpolated for EC-EARTH.

For the creation of the evaluation plots, the signals from the different datasets was calculated over all grid cells where data was available as most have a sufficient amount of grid cells in each latitudinal band. Each dataset was retained at its original resolution for the calculation of the latitudinal averages in order to avoid interpolation errors. The observational data could be





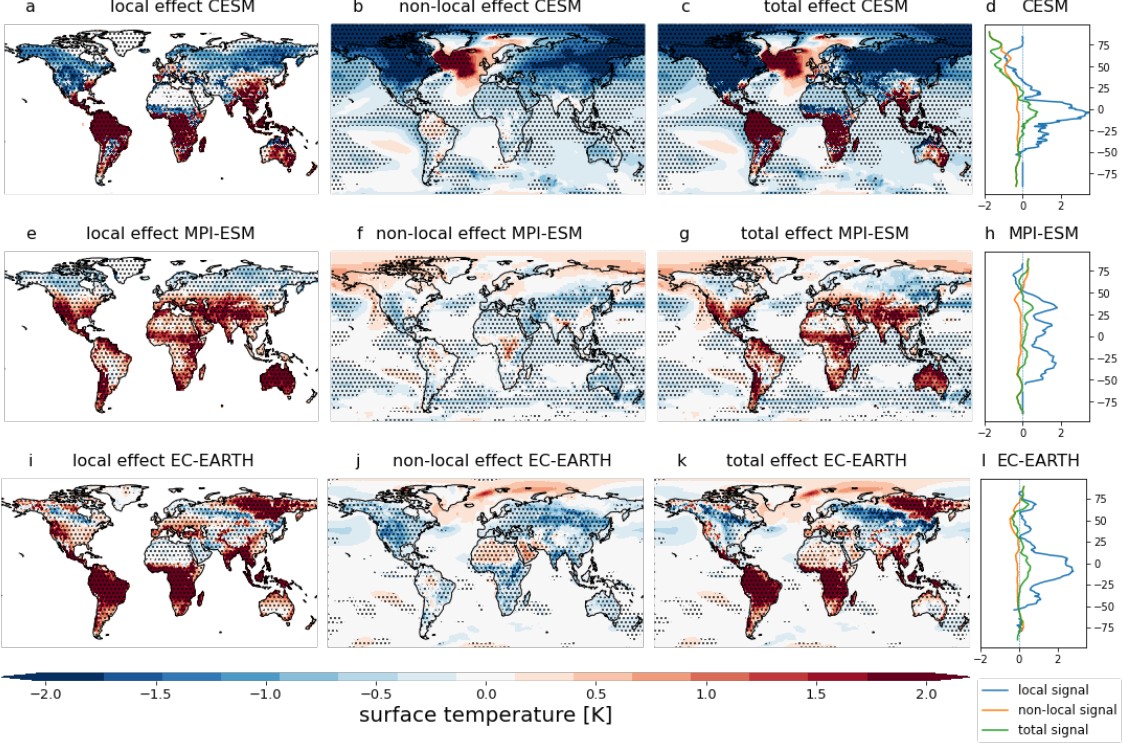

**Figure C3.** Annual mean surface temperature response to full idealised deforestation (CROP-FRST) of CESM, MPI-ESM and EC-EARTH. The local effect in CESM (a), the non-local effect (b) and the total effect (c). The latitudinal average of the local (blue), non-local (yellow) and total (green) signals of CESM (d). (e-h): same as (a-d), but for MPI-ESM. (i-l): same as (a-d), but for EC-EARTH. The stippling on the maps shows grid cells where all 5 ensemble members agree on the sign of change.

directly compared to the output from the CROP-FRST signal separated data as in most grid cells almost a full deforestation occurs as is shown in Figure C2. The corresponding maps showing the local, non-local and total surface temperature effects

are shown in Figure C3.

## Appendix D: Signal separated albedo response

The albedo response (local, non-local and total) are shown for the CROP-FRST case in Figure D1. This clearly illustrates a peculiar feature related to the EC-EARTH model, while albedo change is mainly local (as is the case for MPI-ESM and CESM) it is completely non-local for EC-EARTH. The colorbar range was chosen to clearly show all (even small) changes in albedo.

It shows that the albedo change has a dominant local component for CESM and a smaller non-local component, MPI-ESM only shows a local contribution with no non-local effect and EC-EARTH only shows a non-local contribution.



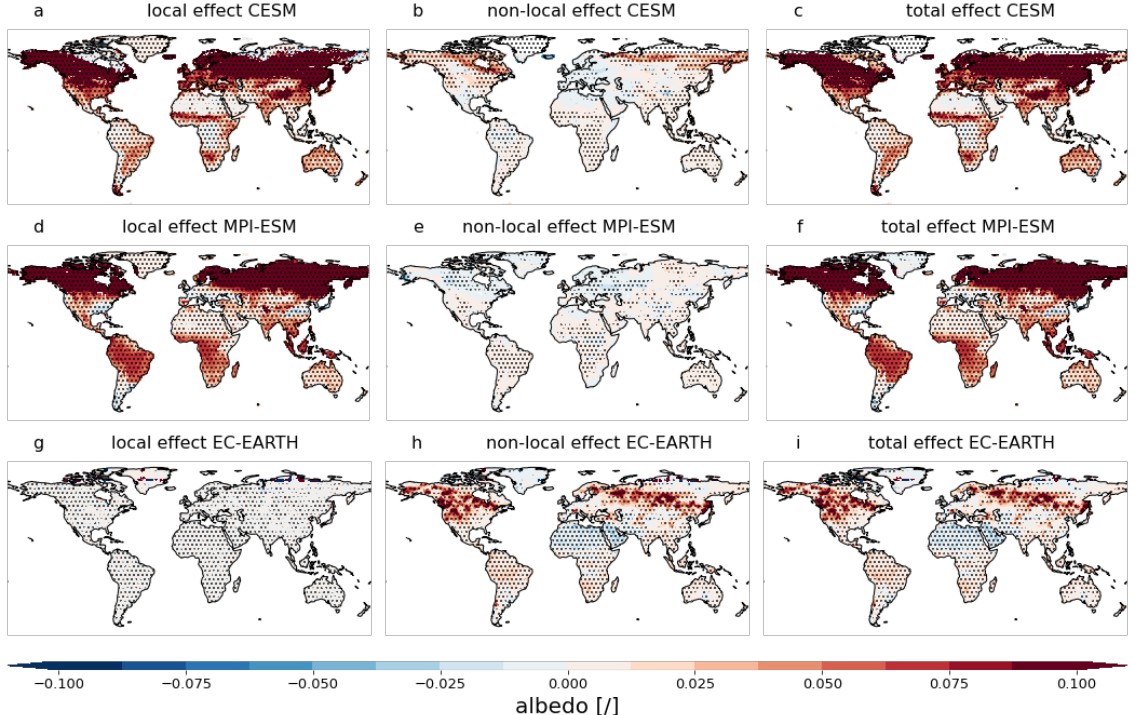

**Figure D1.** Annual mean albedo response to full idealised deforestation (CROP-FRST) of CESM, MPI-ESM and EC-EARTH. The local effect in CESM (a), the non-local effect (b) and the total effect (c). (d-f): same as (a-c), but for MPI-ESM. (g-i): same as (a-c), but for EC-EARTH. The stippling on the maps shows grid cells where all 5 ensemble members agree on the sign of change.

This is further illustrated by Figure D2 where the latitudinal averages of the local, non-local and total effects are compared to the observational datasets from Duveiller et al. (2020) and Li et al. (2015). This again illustrates what was mentioned above, i.e. there is no local component of albedo change for EC-EARTH while this is the dominant component for MPI-ESM and CESM.

However it also clearly shows that even when total effects are considered EC-EARTH strongly underestimates albedo change compared to the observational datasets. This is especially important in the boreal latitudes where EC-EARTH does show a slight increase in the NH, however this effect is still less than half as strong as the observational datasets indicate. Moreover it should be noted that the non-local effects of CESM (which is likely an effect from additional snow due to the non-local cooling, see Figure C3) shows a gradual increase towards the poles, the non-local effect in EC-EARTH in contrast shows a

very similar shape to both the local effects from CESM and MPI-ESM as well as the observational datasets. This indicates that the non-local effect in EC-EARTH is related to the areas which have undergone land cover change in contrast to CESM where it is more related to the latitude and snow cover.

It should be noted that EC-EARTH has undergone less land cover change in the CROP-FRST case compared to the other ESMs as the FRST simulation for this ESM showed very little afforestation amounts (see Figure 1), which likely explains the

underestimation of the total albedo effects for this ESM. However, it remains clear that EC-EARTH has an issue in how the



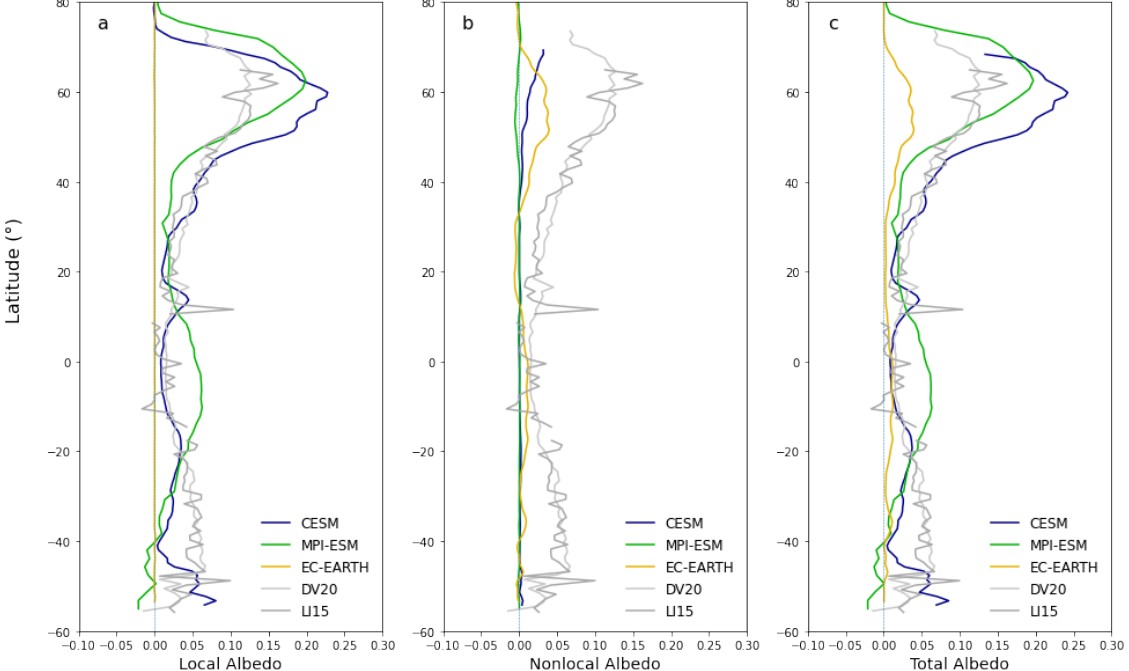

**Figure D2.** Latitudinal evaluation of annual mean albedo derived from full deforestation experiments (CROP-FRST) for CESM (blue), MPI-ESM (green) and EC-EARTH (yellow) with only the local effect shown in panel a, only the non-local effect in panel b, and the total effect in panel c. Note that for all ESMs a running latitudinal mean of 2° was computed. The observational data is shown in grey colours as a reference (Li et al., 2015; Duveiller et al., 2020).

effects on albedo as a consequence to land cover changes is modelled as this should be local by design. This issue should be taken into account within the future development of this ESM as albedo is a crucial variable to understand the effects of land cover changes on the climate.

**Appendix E:  Signal separated response of turbulent heat fluxes, albedo and cloud cover for the different LCLM**




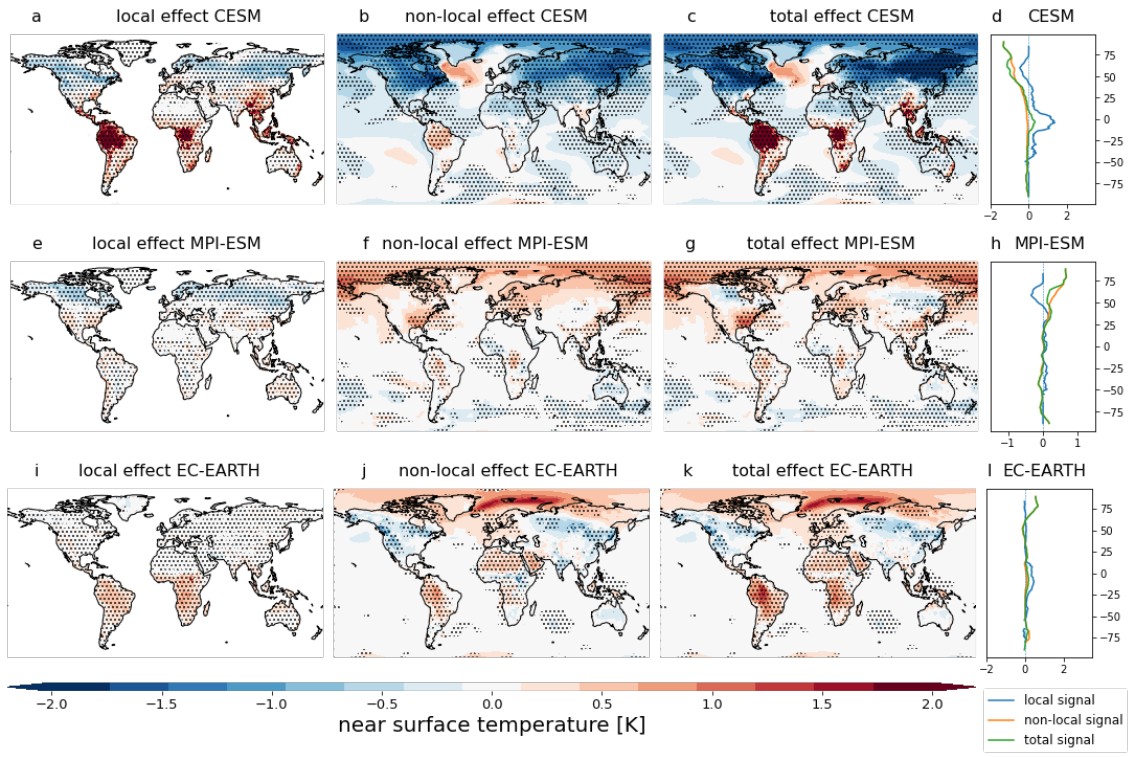

**Figure E1.** Annual mean near-surface temperature response to cropland expansion (CROP-CTL) of CESM, MPI-ESM and EC-EARTH. The local effect in CESM (a), the non-local effect (b) and the total effect (c). The latitudinal average of the local (blue), non-local (yellow) and total (green) signals of CESM (d). (e-h): same as (a-d), but for MPI-ESM. (i-l): same as (a-d), but for EC-EARTH. The stippling on the maps shows grid cells where all 5 ensemble members agree on the sign of change.



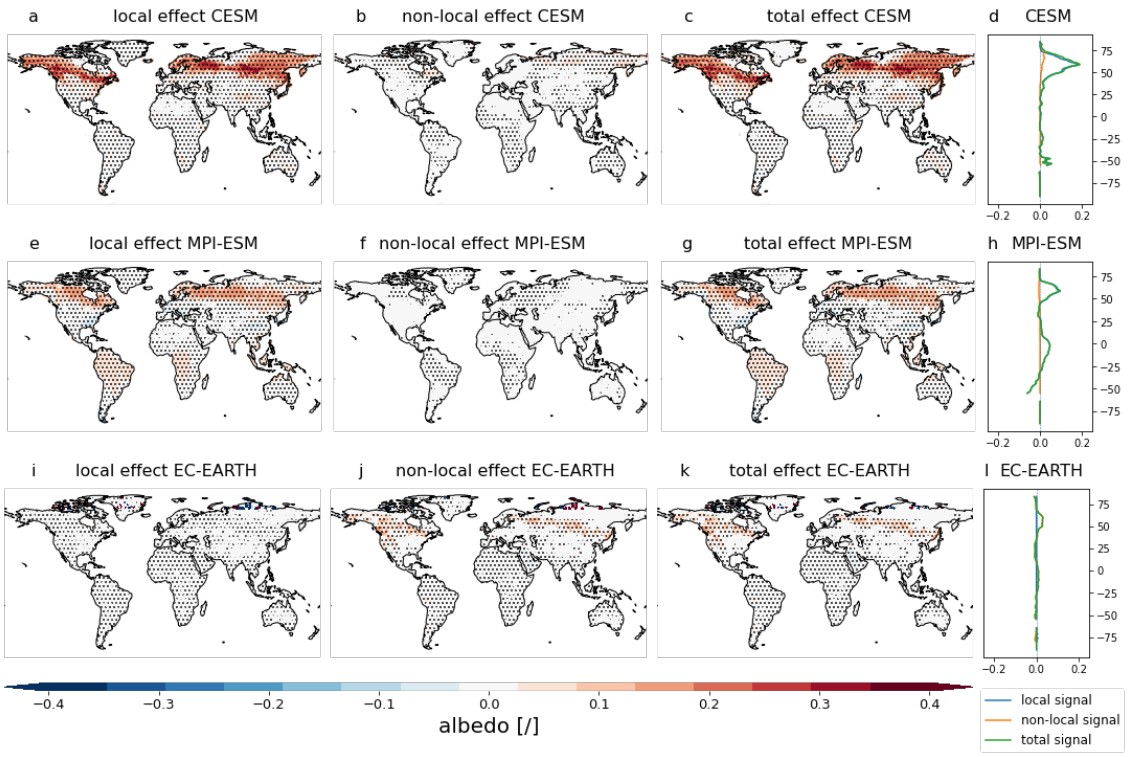

**Figure E2.** Annual mean albedo response to cropland expansion (CROP-CTL) of CESM, MPI-ESM and EC-EARTH. The local effect in CESM (a), the non-local effect (b) and the total effect (c). The latitudinal average of the local (blue), non-local (yellow) and total (green) signals of CESM (d). (e-h): same as (a-d), but for MPI-ESM. (i-l): same as (a-d), but for EC-EARTH. The stippling on the maps shows grid cells where all 5 ensemble members agree on the sign of change.



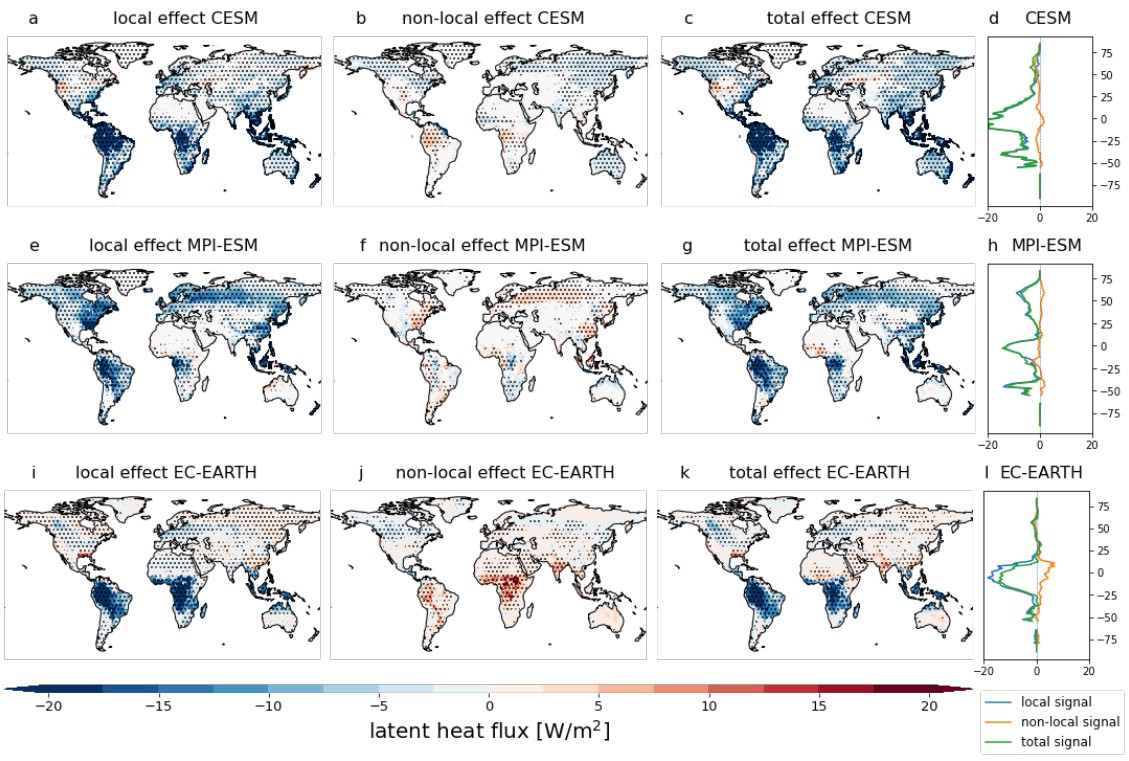

**Figure E3.** Annual mean latent heat flux response to cropland expansion (CROP-CTL) of CESM, MPI-ESM and EC-EARTH. The local effect in CESM (a), the non-local effect (b) and the total effect (c). The latitudinal average of the local (blue), non-local (yellow) and total (green) signals of CESM (d). (e-h): same as (a-d), but for MPI-ESM. (i-l): same as (a-d), but for EC-EARTH. The stippling on the maps shows grid cells where all 5 ensemble members agree on the sign of change.



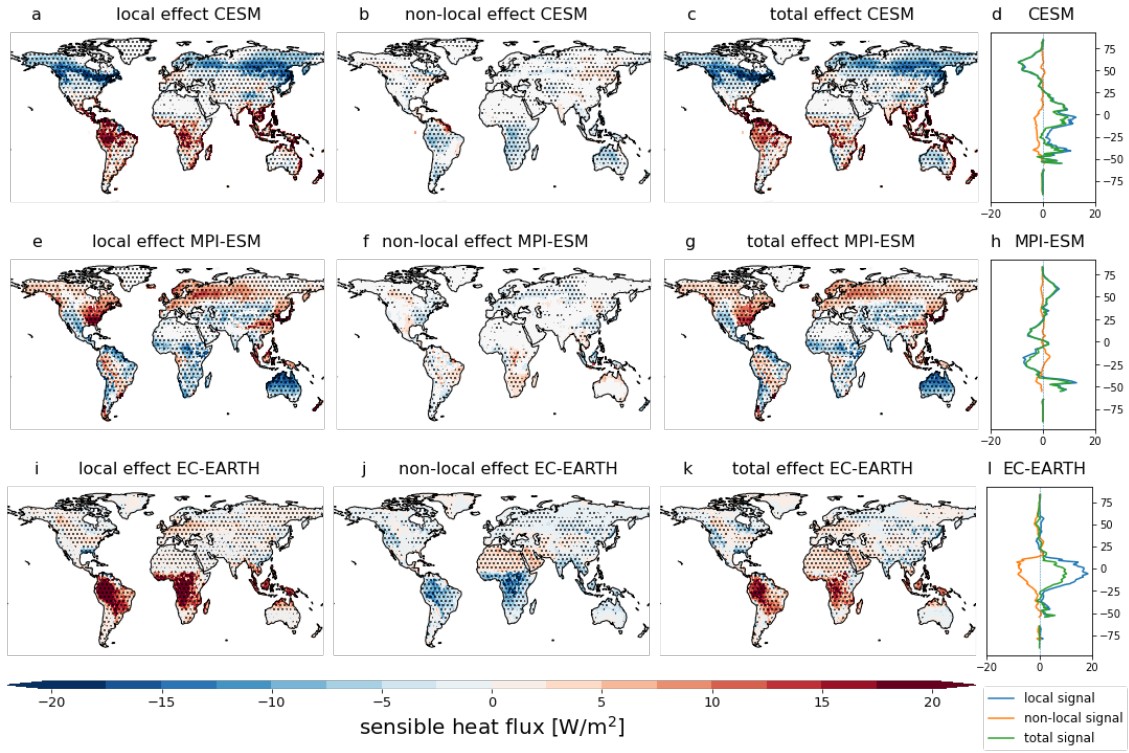

**Figure E4.** Annual mean sensible heat flux response to cropland expansion (CROP-CTL) of CESM, MPI-ESM and EC-EARTH. The local effect in CESM (a), the non-local effect (b) and the total effect (c). The latitudinal average of the local (blue), non-local (yellow) and total (green) signals of CESM (d). (e-h): same as (a-d), but for MPI-ESM. (i-l): same as (a-d), but for EC-EARTH. The stippling on the maps shows grid cells where all 5 ensemble members agree on the sign of change.



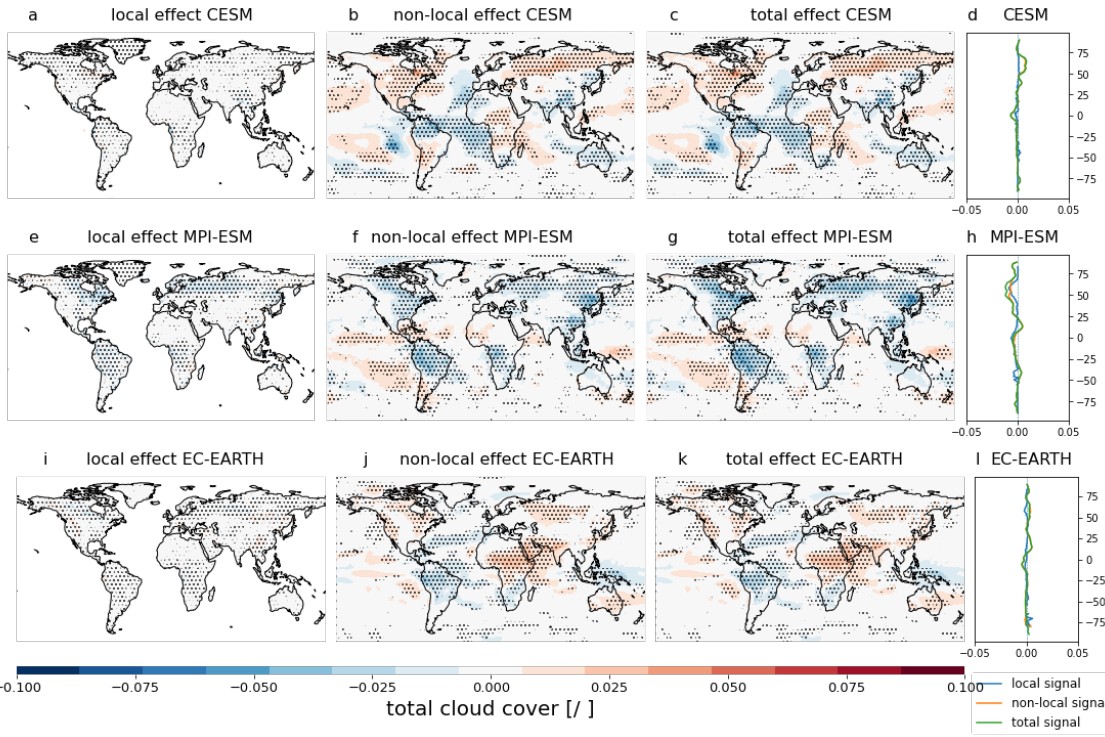

**Figure E5.** Annual mean cloud cover response to cropland expansion (CROP-CTL) of CESM, MPI-ESM and EC-EARTH. The local effect in CESM (a), the non-local effect (b) and the total effect (c). The latitudinal average of the local (blue), non-local (yellow) and total (green) signals of CESM (d). (e-h): same as (a-d), but for MPI-ESM. (i-l): same as (a-d), but for EC-EARTH. The stippling on the maps shows grid cells where all 5 ensemble members agree on the sign of change.



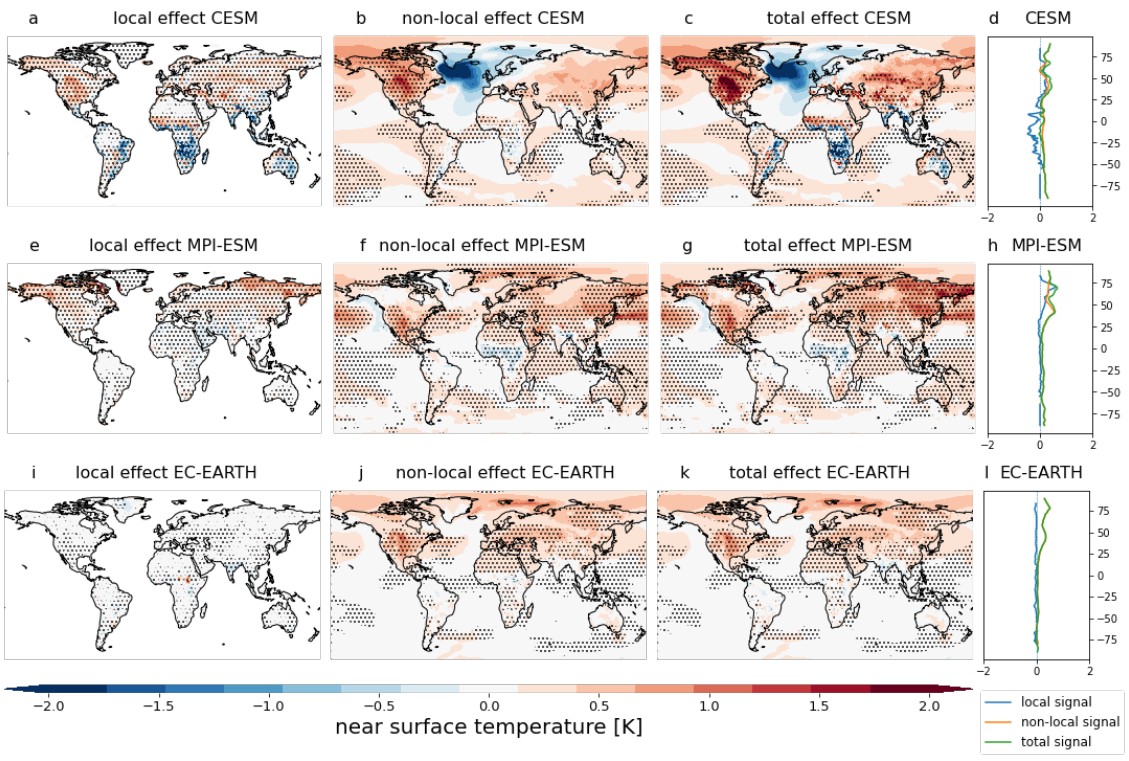

**Figure E6.** Annual mean near-surface temperature response to afforestation (FRST-CTL) of CESM, MPI-ESM and EC-EARTH. The local effect in CESM (a), the non-local effect (b) and the total effect (c). The latitudinal average of the local (blue), non-local (yellow) and total (green) signals of CESM (d). (e-h): same as (a-d), but for MPI-ESM. (i-l): same as (a-d), but for EC-EARTH. The stippling on the maps shows grid cells where all 5 ensemble members agree on the sign of change.



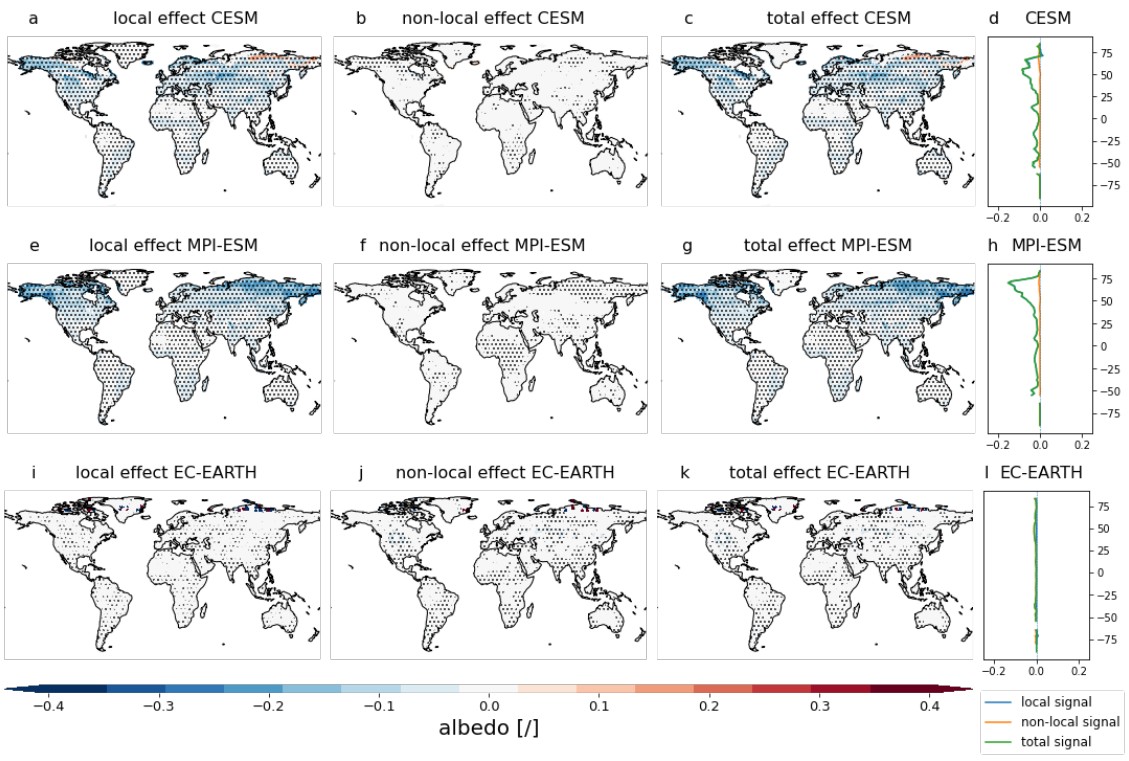

**Figure E7.** Annual mean albedo response to afforestation (FRST-CTL) of CESM, MPI-ESM and EC-EARTH. The local effect in CESM (a), the non-local effect (b) and the total effect (c). The latitudinal average of the local (blue), non-local (yellow) and total (green) signals of CESM (d). (e-h): same as (a-d), but for MPI-ESM. (i-l): same as (a-d), but for EC-EARTH. The stippling on the maps shows grid cells where all 5 ensemble members agree on the sign of change.

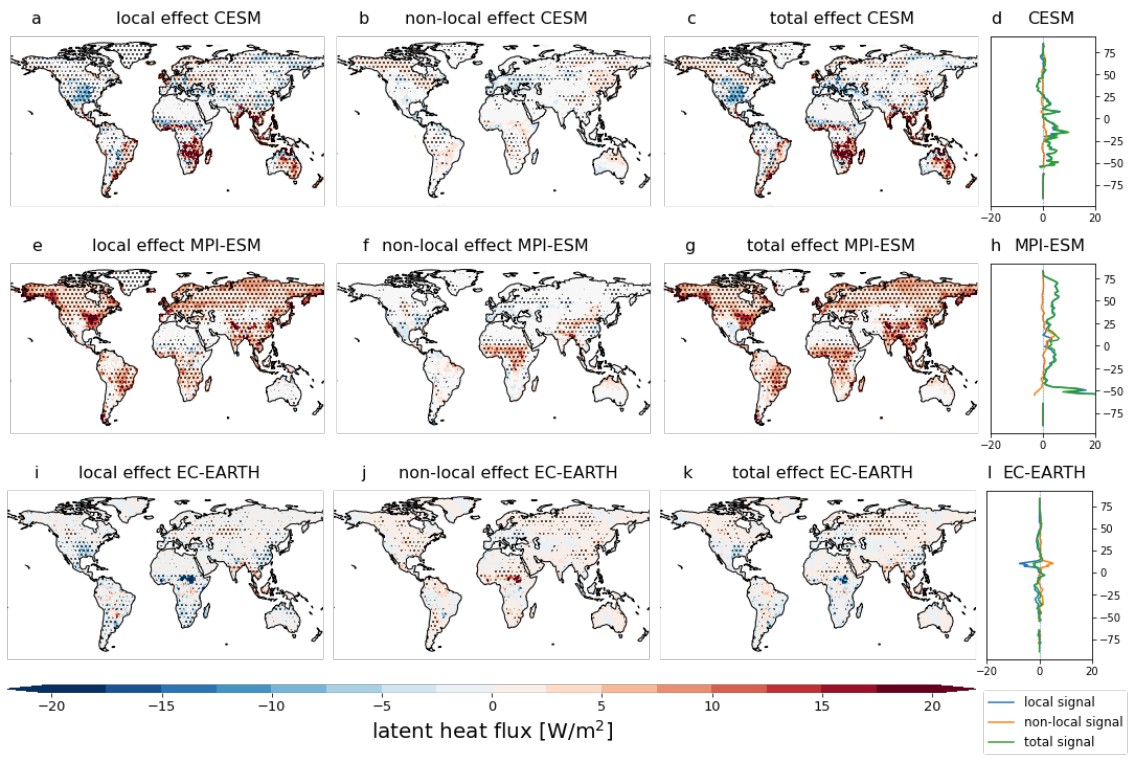

**Figure E8.** Annual mean latent heat flux response to afforestation (FRST-CTL) of CESM, MPI-ESM and EC-EARTH. The local effect in CESM (a), the non-local effect (b) and the total effect (c). The latitudinal average of the local (blue), non-local (yellow) and total (green) signals of CESM (d). (e-h): same as (a-d), but for MPI-ESM. (i-l): same as (a-d), but for EC-EARTH. The stippling on the maps shows grid cells where all 5 ensemble members agree on the sign of change.

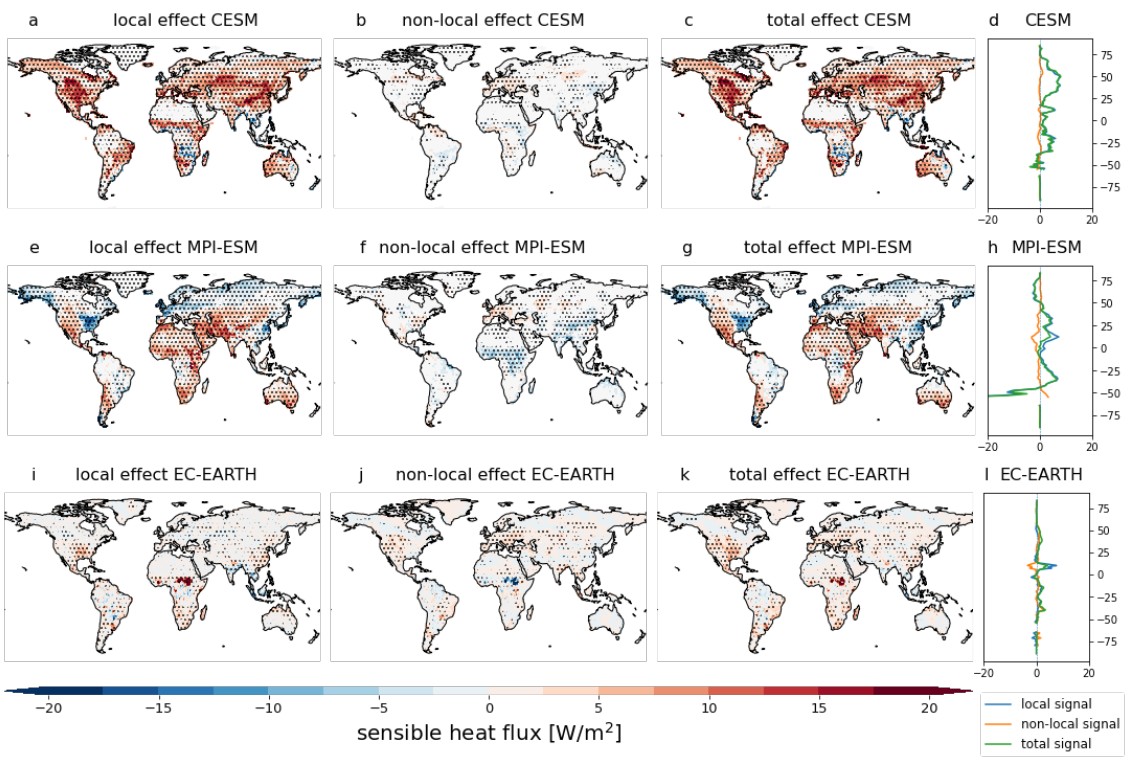

**Figure E9.** Annual mean sensible heat flux response to afforestation (FRST-CTL) of CESM, MPI-ESM and EC-EARTH. The local effect in CESM (a), the non-local effect (b) and the total effect (c). The latitudinal average of the local (blue), non-local (yellow) and total (green) signals of CESM (d). (e-h): same as (a-d), but for MPI-ESM. (i-l): same as (a-d), but for EC-EARTH. The stippling on the maps shows grid cells where all 5 ensemble members agree on the sign of change.



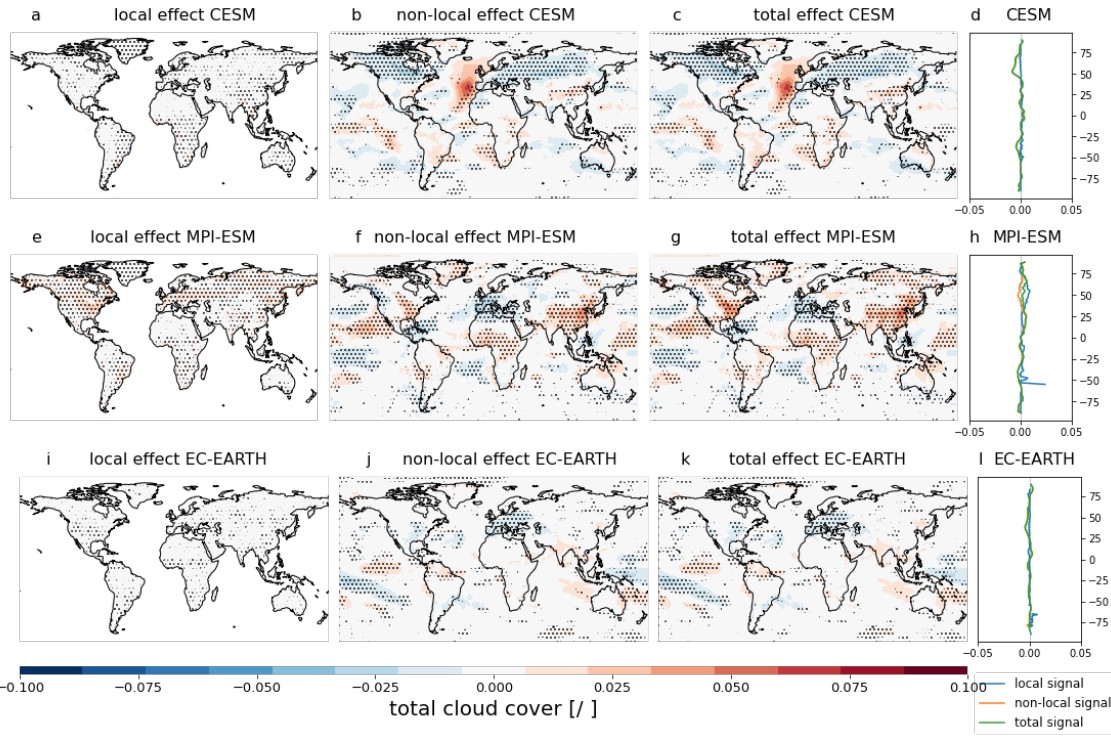

**Figure E10.** Annual mean cloud cover response to afforestation (FRST-CTL) of CESM, MPI-ESM and EC-EARTH. The local effect in CESM (a), the non-local effect (b) and the total effect (c). The latitudinal average of the local (blue), non-local (yellow) and total (green) signals of CESM (d). (e-h): same as (a-d), but for MPI-ESM. (i-l): same as (a-d), but for EC-EARTH. The stippling on the maps shows grid cells where all 5 ensemble members agree on the sign of change.



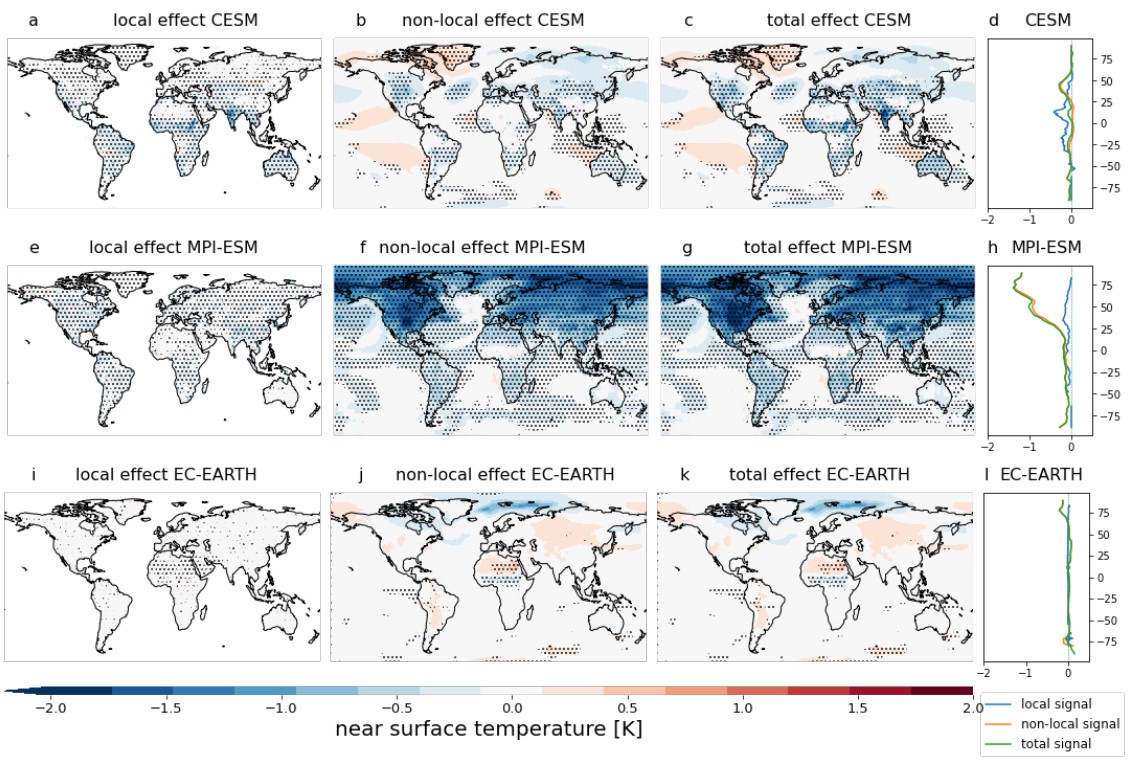

**Figure E11.** Annual mean near-surface temperature response to irrigation expansion (IRR-CROP) of CESM, MPI-ESM and EC-EARTH. The local effect in CESM (a), the non-local effect (b) and the total effect (c). The latitudinal average of the local (blue), non-local (yellow) and total (green) signals of CESM (d). (e-h): same as (a-d), but for MPI-ESM. (i-l): same as (a-d), but for EC-EARTH. The stippling on the maps shows grid cells where all 5 ensemble members agree on the sign of change.



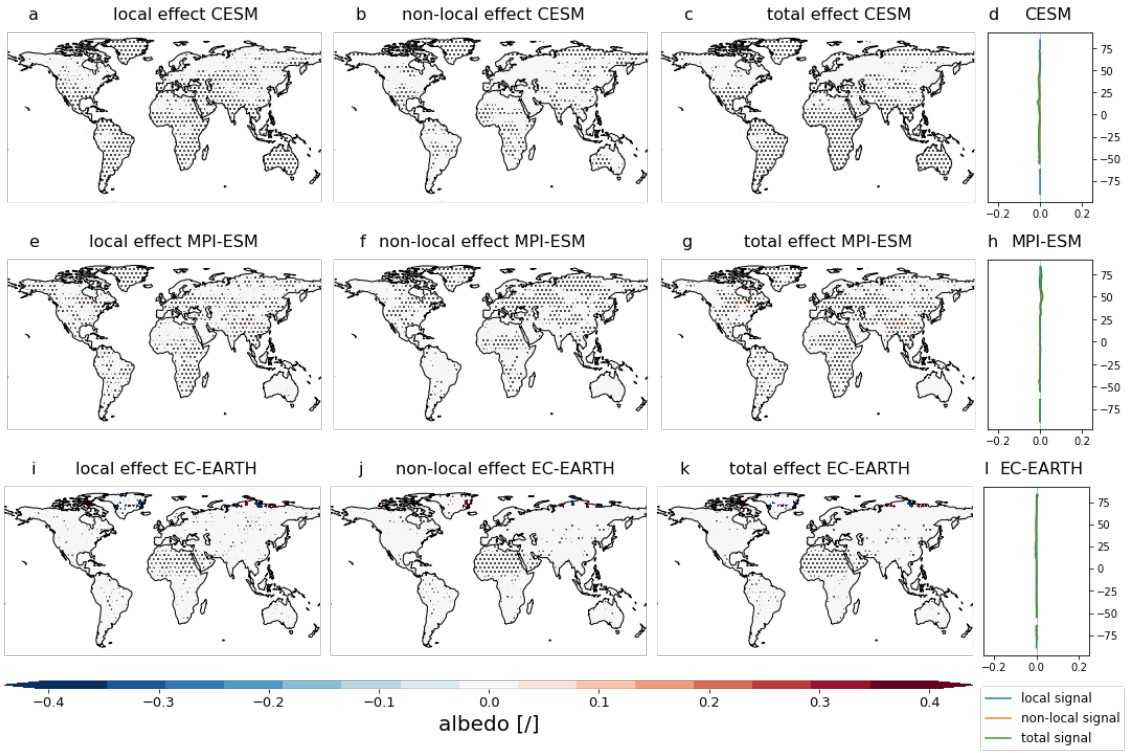

**Figure E12.** Annual mean albedo response to irrigation expansion (IRR-CROP) of CESM, MPI-ESM and EC-EARTH. The local effect in CESM (a), the non-local effect (b) and the total effect (c). The latitudinal average of the local (blue), non-local (yellow) and total (green) signals of CESM (d). (e-h): same as (a-d), but for MPI-ESM. (i-l): same as (a-d), but for EC-EARTH. The stippling on the maps shows grid cells where all 5 ensemble members agree on the sign of change.



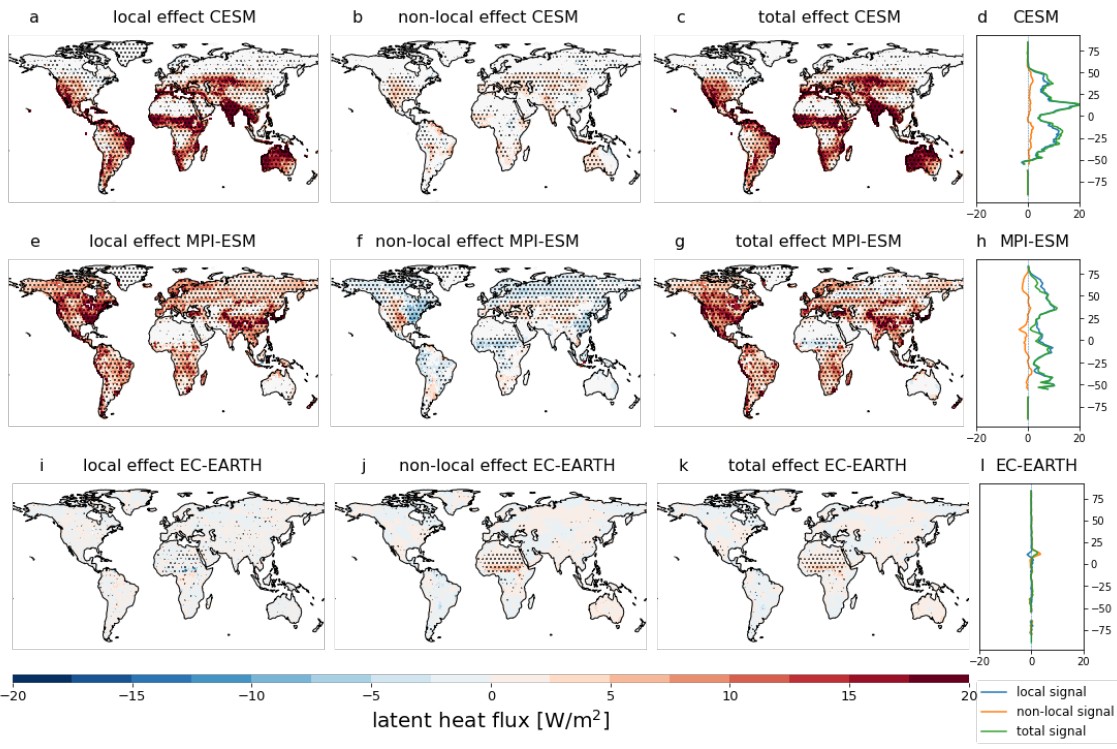

**Figure E13.** Annual mean latent heat flux response to irrigation expansion (IRR-CROP) of CESM, MPI-ESM and EC-EARTH. The local effect in CESM (a), the non-local effect (b) and the total effect (c). The latitudinal average of the local (blue), non-local (yellow) and total (green) signals of CESM (d). (e-h): same as (a-d), but for MPI-ESM. (i-l): same as (a-d), but for EC-EARTH. The stippling on the maps shows grid cells where all 5 ensemble members agree on the sign of change.



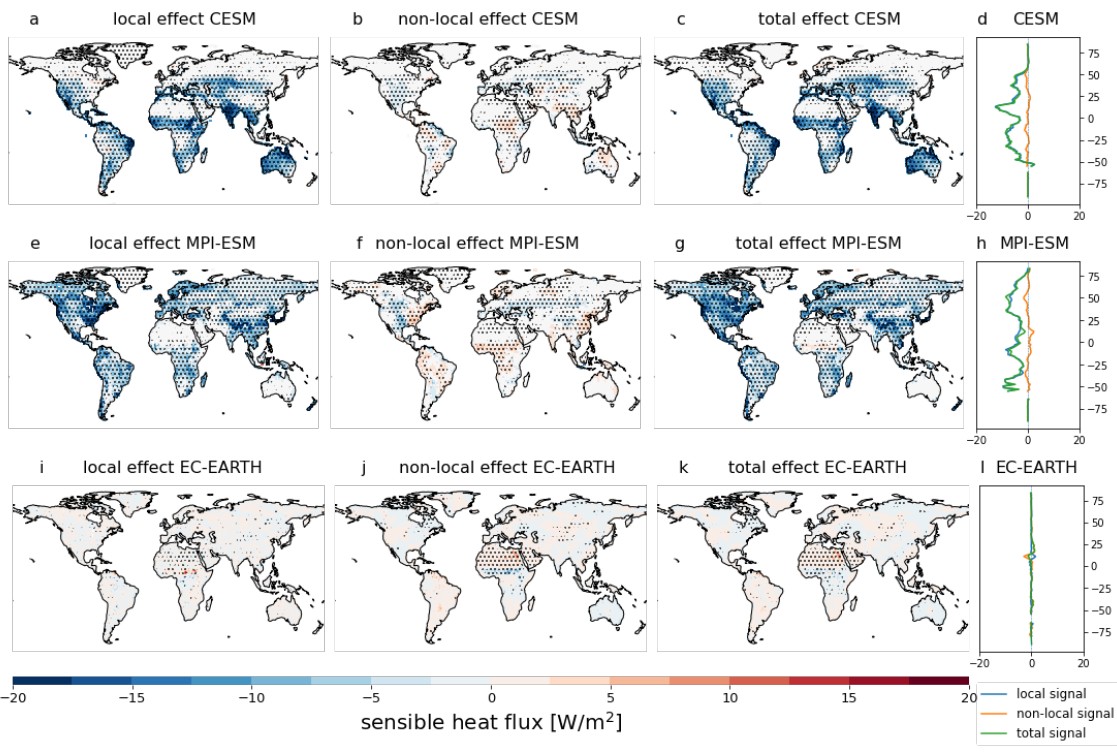

**Figure E14.** Annual mean sensible heat flux response to irrigation expansion (IRR-CROP) of CESM, MPI-ESM and EC-EARTH. The local effect in CESM (a), the non-local effect (b) and the total effect (c). The latitudinal average of the local (blue), non-local (yellow) and total (green) signals of CESM (d). (e-h): same as (a-d), but for MPI-ESM. (i-l): same as (a-d), but for EC-EARTH. The stippling on the maps shows grid cells where all 5 ensemble members agree on the sign of change.



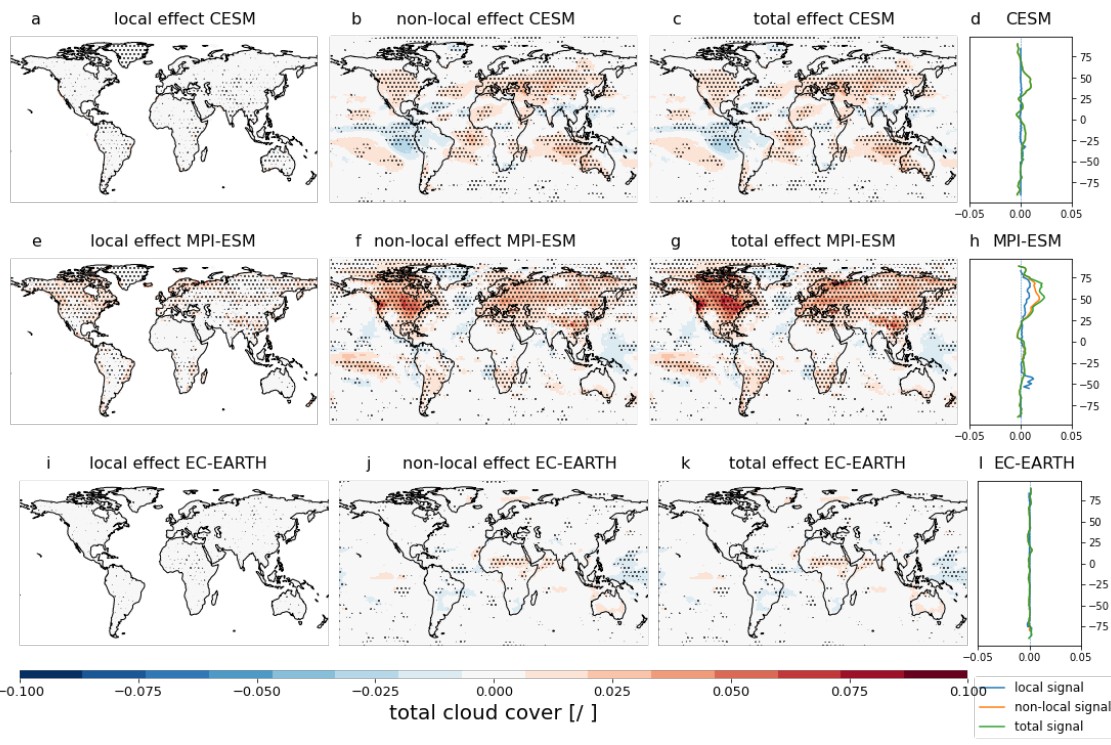

**Figure E15.** Annual mean cloud cover response to irrigation expansion (IRR-CROP) of CESM, MPI-ESM and EC-EARTH. The local effect in CESM (a), the non-local effect (b) and the total effect (c). The latitudinal average of the local (blue), non-local (yellow) and total (green) signals of CESM (d). (e-h): same as (a-d), but for MPI-ESM. (i-l): same as (a-d), but for EC-EARTH. The stippling on the maps shows grid cells where all 5 ensemble members agree on the sign of change.



*Author contributions.* C.F.S., Q.L., W.T., D.C., J.P., E.L.D., S.I.S., F.H., I.M., S.G. and S.D.H. designed the simulation protocol. S.G. and F.H. performed the simulations with MPI-ESM. I.M. performed the simulations with EC-EARTH. S.D.H. performed the simulations with CESM, the data analysis and wrote the paper. I.V. assisted with the setting up of CESM simulations and data analysis. F.H. and S.D.H. performed the post processing for the signal separation. F.L. prepared the EC-EARTH data for post processing and helped with the signal separation for EC-EARTH. E.L.D. helped with the preparation of the land cover data sets. G.D. assisted with the model evaluation. All

authors commented on the paper and provided feedback throughout the data analysis.

*Competing interests.* The authors declare that they have no conflict of interest.

*Acknowledgements.* This work was funded by the DLR/BMBF (DE, Grant No. 01LS1905A), NWO (NL), Belgian Science policy Office (BELSPO) and co-funded by the European Union through the project "LAnd MAnagement for CLImate Mitigation and Adaptation" (LAMACLIMA) (Grant agreement No. 300478), which is part of ERA4CS, an ERA-NET initiated by JPI Climate. I.V. is a research fellow

at the Research Foundation Flanders (FWOTM920). G.D. was supported by the European Research Council (ERC) Synergy Grant "Understanding and Modelling the Earth System with Machine Learning (USMILE)" under Grant Agreement No 855187. The computational resources and services used in this work for the simulations and storage of CESM data were provided by the VSC (Flemish Supercomputer Center), funded by the Research Foundation - Flanders (FWO) and the Flemish Government – department EWI. For the storage of signal separated results and the simulations of MPI-ESM, this work used resources of the Deutsches Klimarechenzentrum (DKRZ) granted by its

Scientific Steering Committee (WLA) under project ID bm1147. F.L. and D.C. acknowledge VIDI-award from Netherlands Organization for Scientific Research (NWO) (Persistent Summer Extremes "PERSIST" project: 016.Vidi.171.011). F.L. would like to thank Philippe Le Sager, Lars Nieradzik and Thomas Reerink for their help in the discussions for the post-processing and interpretations of EC-EARTH Model output. All the simulations from EC-EARTH were carried out on European Center for Medium Range Weather Forecast (ECMWF) platforms. The authors would like to thank Johannes Winckler, Lars Nieradzik, Paul Miller, David Wårlind and the reviewers for their constructive and

useful feedback which has greatly helped improve the manuscript during the review process.



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
