# Peer review of "The biogeophysical effects of idealized land cover and land management changes in Earth System Models"

_EGUsphere, 2023_

## Author Comment (AC1)

**Corrigendum to**
**"The biogeophysical effects of idealized land cover and land management changes in Earth System Models" published in Earth Syst. Dynam., 0, 1–11, 2023**

**Steven J. De Hertog**[1]**, Felix Havermann**[2]**, Inne Vanderkelen**[1]**, Suqi Guo**[2]**, Fei Luo**[3,4]**, Iris Manola**[3]**, Dim Coumou**[3,4]**, Edouard L. Davin**[5,6,7]**, Gregory Duveiller**[8]**, Quentin Lejeune**[9]**, Julia Pongratz**[2,10]**, Carl-Friedrich Schleussner**[9]**, Sonia I. Seneviratne**[11]**, and Wim Thiery**[1]

[1]Vrije Universiteit Brussel, Department of Hydrology and Hydraulic Engineering, Brussels, Belgium
[2]Ludwig-Maximilians-University Munich, Department of Geography, Munich, Germany
[3]Vrije Universiteit Amsterdam, Institute for Environmental studies, Amsterdam, Netherlands
[4]Royal Netherlands Meteorological Institute (KNMI), De Bilt, Netherlands.
[5]Wyss Academy for Nature, University of Bern, Bern, Switzerland.
[6]Climate and Environmental Physics division, University of Bern, Bern, Switzerland.
[7]Oeschger Centre for Climate Change Research, University of Bern, Bern, Switzerland
[8]Max-Planck-Institute for Biogeochemistry, Jena, Germany
[9]Climate Analytics, Berlin, Germany
[10]Max Planck Institute for Meteorology, Hamburg, Germany
[11]ETH Zurich, Institute for Atmospheric and Climate Science, Zurich, Switzerland.

**Correspondence:** Steven De Hertog (steven.de.hertog@vub.be)

This corrigendum is supposed to correct some mistakes that got into the final publication during the production process of the paper. Due to an oversight these mistakes were only discovered after publication of the paper.

5 Firtsly, the affiliations in the initial article for Edouard Davin were outdated, here below the updated affiliations are given:
1) Wyss Academy for Nature, University of Bern, Bern, Switzerland
2) Climate and Environmental Physics division, University

10 of Bern, Bern, Switzerland
3) Oeschger Centre for Climate Change Research, University of Bern, Bern, Switzerland

Furthermore Figure 5 in the published manuscript is
15 wrong, accidentally the maps for MPI-ESM (2nd row) where showing the values of near-surface temperature instead of surface temperature as the header intended. The authors would like to acknowledge Johannes Winckler for his keen eye on spotting this error.

20

Here below the changes changes required to correct the text (section 3.2.1.) are described.

– line 364: remove lower so sentence becomes: MPI-ESM also simulates local warming over the tropics, but with a different spatial pattern and magnitude compared to 25 CESM and EC-EARTH.

– line 373-174: replace by: In all ESMs, the local signals dominate the total response in the tropics.

Lastly an error was made in the postprocessing of the EC-EARTH output as the model specific sign convention of the 30 turbulent heat fluxes (latent heat and sensible heat) was not taken into account. Therefore all figures showing these turbulent heat fluxes are wrong and should have an inverted sign. In general the main conclusions still hold, the main difference is that EC-EARTH is more in line with the other ESMs 35 (especially for latent heat over the tropics). Regarding the energy balance decomposition latent and sensible heat should be switched. Below we show the relevant figures (i.e. Figure 4 panels a and b, Figure 9, Figure 10 and appendix figures

[Figure]

**Figure 5.** Annual mean surface temperature response to cropland expansion (CROP-CTL) of CESM (top row), MPI-ESM (middle row) and EC-EARTH (bottom row). For CESM: the local effect (a), the non-local effect (b) and the total effect (c), the global latitudinal average of the local (blue), non-local (yellow) and total (green) signals (d). (e-h): same as (a-d), but for MPI-ESM. (i-l): same as (a-d), but for EC-EARTH. The stippling on the maps shows grid cells where all 5 ensemble members agree on the sign of change.

E3, E4, E8, E9, E13, E14) and explain any changes in the accompanying text.

In Section 3.1. the following lines should be changed:

- line 311-312: remove 'whereas EC-EARTH shows opposite patterns' and instead add 'and EC-EARTH' as the results are now in line with the other ESMs.

- line 316-317: Change increase to decrease.

- line 322-323: Change decrease to increase.

In Section 3.3.1. the following line should be changed: Line 435-439: Instead of 'both CESM and MPI-ESM' it should be 'all ESMs'. The entire sentence starting with 'This is in strong contrast to EC-EARTH' should be removed as well as the following sentence which starts with 'This is most likely caused by over productive cropland' as this explanation is wrong.

In section 3.3.2. the following line should be changed: line 467-469: In EC-EARTH, the cooling is caused by changes in sensible heat flux (in stead of latent heat flux) and incoming longwave radiation, but is counteracted by a decrease in latent heat flux (instead of sensible heat flux).

In section 4.1. the messages still hold, however one example is wrong and should be changed: line 512-514 which is 'Both MPI-ESM and CESM show that local latent heat flux changes determine the surface temperature response in the tropics, while in EC-EARTH, a decrease in sensible heat flux along with an increase in incoming longwave radiation induce the warming response.' no longer holds and therefore should be replaced with 'All three ESMs show that local latent heat flux changes determine the surface temperature response in the tropics. However, the role of local sensible heat flux changes differs across ESMs, showing a cooling effect in CESM and EC-EARTH in contrast to MPI-ESM where it has a warming effect.'. Although the meaning of the sentence has changed, it doesn't alter the message as they were just both meant to illustrate the statement on line 511: 'However, they (i.e. the ESMs) disagree on how these changes occur.', which even though the differences are less strong now still holds.

[Figure]

**Figure 4.** Latitudinal evaluation of local energy and climate variables derived from full deforestation experiments (CROP-FRST). The local effect simulated by CESM (blue), MPI-ESM (green) and EC-EARTH (yellow) of latent heat flux (W/m$^2$ ) (a) compared to observational estimates by Li et al. (2015); Duveiller et al. (2018) (DV20 and LI15, respectively), of sensible heat flux (W m$^{-2}$) (b) compared to Duveiller et al. (2018) (DV20), of albedo (-) (c) compared to Li et al. (2015); Duveiller et al. (2018) (LI15 and DV20) and near surface temperature (K) (d) compared to Alkama and Cescatti (2016); Duveiller et al. (2020) (AL16 and DV20). Note that for all ESMs a running latitudinal mean of 2° was computed.

In section 4.4., line 649-650 the following part ('unrealistic response in the turbulent energy fluxes and the') should be removed so the sentence becomes: 'This causes some clear biases such as the unrealistic partition of albedo as a non-local feature in EC-EARTH (Figure 4c).'

In section 5, line 692-693 should be removed, i.e. 'However, the sign of change in the turbulent heat fluxes is opposite in EC-EARTH compared to CESM and MPI-ESM.' The rest of the conclusion holds.
The updated appendix figures are added as a reference.

**References**

Alkama, R. and Cescatti, A.: Climate change: Biophysical climate impacts of recent changes in global forest cover, Science, 351, 600–604, https://doi.org/10.1126/science.aac8083, 2016.

Duveiller, G., Hooker, J., and Cescatti, A.: The mark of vegetation change on Earth's surface energy balance, Nature Communications, 679, https://doi.org/10.1038/s41467-017-02810-8, 2018.

Duveiller, G., Caporaso, L., Abad-Viñas, R., Perugini, L., Grassi, G., Arneth, A., and Cescatti, A.: Local biophysical effects of land use and land cover change: towards an assessment tool for policy makers, Land Use Policy, 91, 104 382, https://doi.org/10.1016/j.landusepol.2019.104382, 2020.

Li, Y., Zhao, M., Motesharrei, S., Mu, Q., Kalnay, E., and Li, S.: Local cooling and warming effects of forests based on satellite observations, Nature Communications, 6, https://doi.org/10.1038/ncomms7603, 2015.

[Figure]

**Figure 9.** The energy balance decomposition of the local surface temperature for the different latitudinal bands. The response to cropland expansion (CROP-CTL) for CESM (a), MPI-ESM (b), and EC-EARTH (c), the response to afforestation (FRST-CTL) for CESM (d), MPI-ESM (e), and EC-EARTH (f) and the response to irrigation expansion (IRR-CROP) for CESM (g) and MPI-ESM (h). EC-EARTH is not shown for irrigation expansion as the local effects are too small for any meaningful analysis.

[Figure]

**Figure 10.** Global average seasonal cycle of energy balance decomposition of local surface temperature. The response to cropland expansion (CROP-CTL) for CESM (a), MPI-ESM (b), and EC-EARTH (c), the response to afforestation (FRST-CTL) for CESM (d), MPI-ESM (e), and EC-EARTH (f) and the response to irrigation expansion (IRR-CROP) for CESM (g) and MPI-ESM (h).

[Figure]

**Figure E3.** Annual mean latent heat flux response to cropland expansion (CROP-CTL) of CESM, MPI-ESM and EC-EARTH. The local effect in CESM (a), the non-local effect (b) and the total effect (c). The latitudinal average of the local (blue), non-local (yellow) and total (green) signals of CESM (d). (e-h): same as (a-d), but for MPI-ESM. (i-l): same as (a-d), but for EC-EARTH. The stippling on the maps shows grid cells where all 5 ensemble members agree on the sign of change.

[Figure]

**Figure E4.** Annual mean sensible heat flux response to cropland expansion (CROP-CTL) of CESM, MPI-ESM and EC-EARTH. The local effect in CESM (a), the non-local effect (b) and the total effect (c). The latitudinal average of the local (blue), non-local (yellow) and total (green) signals of CESM (d). (e-h): same as (a-d), but for MPI-ESM. (i-l): same as (a-d), but for EC-EARTH. The stippling on the maps shows grid cells where all 5 ensemble members agree on the sign of change.

[Figure]

**Figure E7.** Annual mean latent heat flux response to afforestation (FRST-CTL) of CESM, MPI-ESM and EC-EARTH. The local effect in CESM (a), the non-local effect (b) and the total effect (c). The latitudinal average of the local (blue), non-local (yellow) and total (green) signals of CESM (d). (e-h): same as (a-d), but for MPI-ESM. (i-l): same as (a-d), but for EC-EARTH. The stippling on the maps shows grid cells where all 5 ensemble members agree on the sign of change.

[Figure]

**Figure E8.** Annual mean sensible heat flux response to afforestation (FRST-CTL) of CESM, MPI-ESM and EC-EARTH. The local effect in CESM (a), the non-local effect (b) and the total effect (c). The latitudinal average of the local (blue), non-local (yellow) and total (green) signals of CESM (d). (e-h): same as (a-d), but for MPI-ESM. (i-l): same as (a-d), but for EC-EARTH. The stippling on the maps shows grid cells where all 5 ensemble members agree on the sign of change.

[Figure]

**Figure E13.** Annual mean latent heat flux response to irrigation expansion (IRR-CROP) of CESM, MPI-ESM and EC-EARTH. The local effect in CESM (a), the non-local effect (b) and the total effect (c). The latitudinal average of the local (blue), non-local (yellow) and total (green) signals of CESM (d). (e-h): same as (a-d), but for MPI-ESM. (i-l): same as (a-d), but for EC-EARTH. The stippling on the maps shows grid cells where all 5 ensemble members agree on the sign of change.

[Figure]

**Figure E14.** Annual mean sensible heat flux response to irrigation expansion (IRR-CROP) of CESM, MPI-ESM and EC-EARTH. The local effect in CESM (a), the non-local effect (b) and the total effect (c). The latitudinal average of the local (blue), non-local (yellow) and total (green) signals of CESM (d). (e-h): same as (a-d), but for MPI-ESM. (i-l): same as (a-d), but for EC-EARTH. The stippling on the maps shows grid cells where all 5 ensemble members agree on the sign of change.

---

## Author Comment (AC2)

**The biogeophysical effects of idealized land cover and land management changes in earth system models**

*Response to reviewers*

*13/04/2023*

We would like to thank all reviewers for their dedicated time reviewing the manuscript and for their useful and constructive suggestions. We especially thank reviewer 2 for highlighting mistakes and misinterpretation related to different modelling philosophies of EC-EARTH compared to the other two ESMs. We believe the review phase has greatly benefitted the scientific quality of the manuscript. As reviewer 1 did not have any additional comments we focus on the comments of reviewer 2 here.

| Reviewer 2 Comment 1 |
| --- |
| L126 Update Döscher et al. to the published article. |

**Response**

We thank the reviewer for pointing this out, the manuscript now refers to the published article.

L186-195 This is not totally true. EC-Earth has a dynamic vegetation model (LPJ-GUESS, Smith et al. 2014) and this is different from CESM and MPI-ESM. The main difference is in the assumption of what a forest is. For CESM and MPI-ESM the areas that are afforested are assumed to be a forest directly (in a physical sense) despite that the tree PFTs don't have any biomass. The forest just appears directly after the land cover map is changed. This is an assumption/model setup that you can have, but why this was made differently for EC-Earth I don't understand. In the EC-Earth setup of this study, it is the biomass of the trees that determines the physical parameters of the forest which is the opposite of the other models. It will take time for a tree to grow to a size that affects anything. And perhaps in many of the areas where the forest should be, the tree PFTs might not be productive and no tree biomass is created. I guess that the tree biomass between the three models is similar in areas of afforestation but definitely, the physical parameters are different due to the differences in model setups. This difference in the setup is what drives the difference between the models, not that EC-Earth has a dynamic vegetation model. To state that the same approach isn't possible for EC-Earth (having a physical forest directly after the change of land cover map) is not true. There is no switch for this, but a very small hack of the code would make the EC-Earth behave in the same way as CESM and MPI-ESM. It is pity that this hasn't been done as now any FRST-CTRL comparison between the models is extremely hard. This also explains the albedo discrepancy or EC-Earth compared to the other models in Figure 4c and D2 and makes your statement in L778-783 wrong. There isn't an error in how EC-Earth represents albedo, but an error in your model setup, as explained above. If there isn't any extensive new tree biomass in afforested areas, then you won't see a change in albedo. The other models see a change in albedo due to the physical parameters being set to be a forest, hence they see a large change in albedo. This discrepancy between the model setups needs to be made very clear, as the author seems to not understand it themselves and make wrong statements about EC-Earth due to their error in setting up the model simulations. Also, all other FRST-CTRL comparisons with EC-Earth show very small effects. This is also due to the above-explained setup error. And FRST-CROP comparisons with EC-Earth will mainly be driven by the CROP change to CTRL as FRST to CTRL shows so minor differences. So the effect of the different model setups needs to be made much clearer.

**Response**

We thank the reviewer for his comment and clear explanation. This simulation setup was chosen as we were not aware of the possibility of a workaround to prevent dynamic vegetation within EC-EARTH which would allow us to force this model with prescribed land cover maps. If we would have been

aware of this option we would have definitely chosen for this option as this would indeed improve the comparability of the FRST simulation. We agree with the reviewer that this should be made more explicit within the manuscript. The link to the issue of lack of local albedo was indeed not interpreted as such but we thank the reviewer for clearly highlighting this here and already providing explanations. In the revised version of the manuscript we have included several clarifications and added these here below, the altered parts are indicated in blue. In the last paragraph of methods section 2.1.1 where the ESMs are presented we made this difference in assumptions explicit:

> There are some important differences in how the different ESMs treat land cover. They have a different amount of PFTs which are also defined in different categories. Moreover while in MPI-ESM and CESM land cover is handled within one single sub-model (their respective land surface schemes JSBACH and CLM) and is prescribed, in EC-EARTH there are different models for vegetation dynamics and biogeochemistry (LPJ-GUESS) and for the water and energy cycle (HTESSEL). This implies that for CESM and MPI-ESM the areas which are afforested are assumed to be a physical forest immediately. This is in contrast to EC-EARTH where the dynamic vegetation model determines the physical properties of trees from biomass buildup through vegetation growth.

Next we highlighted this again in the results section within the 'Evaluation of biogeophysical response to deforestation' section (section 3.1) where we make the link with our simulation setup and lack of local albedo explicit:

> In EC-EARTH the local albedo change is zero (Figure 4c), however there is a stronger non-local albedo change  (Figure D1). The non-local albedo change is near-zero except over boreal latitudes, (Figure D2). This could be caused by the differences in simulation setup for EC-EARTH where the forest needs to establish throughout the simulation (e.g., biomass and specific land surface properties such as vegetation roughness length, LAI, albedo) under the local environmental conditions while in CESM and MPI-ESM some of the specific land surface parameters are immediately established. This albedo bias due to differences in simulation setup likely explains the lack of cooling in boreal latitudes for this EC-EARTH (Figure  3).

This link to simulation setup and lack of albedo changes in EC-EARTH is further highlighted in Appendix D which has been largely rewritten to further explain that the lock of forest growth is the reason for the lack of local albedo in EC-EARTH:

> This is further illustrated by Figure D2 where the latitudinal averages of the local, non-local and total effects are compared to the observational datasets from Duveiller et al. (2020) and Li et al. (2015). This again illustrates what was mentioned above, i.e. there is no local component of albedo change for EC-EARTH while this is the dominant component for MPI-ESM and CESM. However it also clearly shows that even when total effects are considered EC-EARTH strongly underestimates albedo change compared to the observational datasets. This is especially important in the boreal latitudes where EC-EARTH does show a slight increase in the NH, however this effect is still less than half as strong as the observational datasets indicate.  Due to the specific simulation setup used in this study EC-EARTH  is not able to grow sufficient amounts of vegetation to cause a clear local albedo effect, only

non-local effects are visible for this ESM. In CESM and MPI-ESM  this issue does not occur as the land cover change  immediately implements a physical forest and the related land surface properties without the need for these to grow.

It should be noted that due to this issue, EC-EARTH has undergone less land cover change in the CROP-FRST case compared to the other ESMs as the FRST simulation for this ESM showed very little afforestation amounts (see Figure 1)~~, which likely explains the underestimation of the total albedo effects for this ESM. However, it remains clear that EC-EARTH has an issue in how the effects on albedo as a consequence to land cover changes is modelled as this should be local by design. This issue should be taken into account within the future development of this ESM as albedo is a crucial variable to understand the effects of land cover changes on the climate.~~ and these forests are only established to a limited extent causing smaller biophysical effects on the atmosphere.

Next, we repeat this explicitly in the first section of the discussion when highlighting some inconsistencies across the ESMs (section 4.1):

Although we have harmonised the land cover and management representation across the different models, strong differences remain, most notably in the implementation of irrigation expansion and afforestation (Figure 1). This implies that the comparison of the different simulations across ESMs is not perfect and inconsistencies can be caused by disparity in model structure and by spatial differences and differences in extent and implementation of the applied LCLMC. As for afforestation, the differences found here were mainly caused by the  differences of implementation of forests in EC-EARTH (where the forest and respective land surface properties change throughout the simulation) compared to CESM and MPI-ESM which start of with a physical forest and its land surface properties.

Finally, we explicitly state the possibility of forcing EC-EARTH to behave like CESM and MPI-ESM (by prescribing land cover) within the last section of the discussion regarding Limitations and Future Outlook (section 4.4):

The afforestation implemented in  EC-EARTH~~, even though it has a highly advanced land model (LPJ-GUESS), the interface with the atmosphere is handled by a more simple submodel (HTESSEL) within the atmosphere model IFS. This causes some clear biases such as the unrealistic partition of albedo as a non-local feature in EC-EARTH (Figure 4c). Addressing these biases could be a useful strategy when further developing this ESM to make land cover induced climate effects more realistic.~~ in this study could have been improved and made more comparable to the other ESMs by changing the simulation setup. For example by forcing forest to exist from the start of the simulation (as was done in MPI-ESM and CESM) in stead of allowing EC-EARTH to model afforestation as default within the dynamic vegetation model LPJ-GUESS.

Reviewer 2 Comment 3

L197-202 The irrigation in EC-Earth is only applied to the dynamic vegetation model. The physical model (IFS/HTESSEL) doesn't see any difference between an irrigated or rainfed cropland as it has a separate water cycle compared to LPJ-GUESS (see Döscher et al. 2022). So, no water is added to the physical model. The only difference it sees is the change in LAI between an irrigated and rainfed crop PFT. This needs to be explained as this makes it easier to explain the small temperature impact in Figure 7i-l. It is too late to bring this up in the discussion (L639-642), it should be explained already when the experimental design is described and hence EC-Earth should be fully excluded from the IRR-CROP analysis as has been done in figures 9 and 10.

**Response**

We thank the reviewer for highlighting this aspect and we agree that this lack of communication between the atmosphere and dynamic vegetation components is important to clearly state early in the manuscript. However, this is already highlighted in the manuscript at several occasions in the current version of the manuscript as shown below:

Appendix B explicitly states this issue regarding EC-EARTH and the appendix is referenced in methods section 2.1.2:

> Although the individual implementations of the irrigation parameterisation differ, all models follow a similar logic. Once a crop suffers a certain amount of water stress (defined differently in the models, see Appendix B), this amount is replenished by applying an irrigation flux until the water stress is relieved

Second this is mentioned in the description of the irrigation induced temperature results shown in Figure 7 in the first paragraph of section 3.2.3:

> In the idealised irrigation expansion sensitivity experiment (IRR-CROP, i.e. irrigation expansion in a full cropland world), both MPI-ESM and CESM agree on the irrigation-induced reduction in local surface temperature, while irrigation expansion in EC-EARTH does not induce any local effects (Figure 7). The very limited local effects in EC-EARTH are caused by a lack of moisture exchange between IFS and LPJ-GUESS, whereby water added in LPJ-GUESS for irrigation does not affect the moisture fluxes in IFS. Hence, in EC-EARTH, irrigation affects crop growth and albedo but does not alter turbulent surface fluxes.

However as it is clear from the reviewers comment that this should be brought forward more clearly we also added an explanation of this issue explicitly in the last paragraph of methods Section 2.1.2 regarding the experimental setup:

> Note that within EC-EARTH, irrigation is implemented in the dynamic vegetation model LPJ-GUESS but not within the atmosphere model IFS (due to both models having a separate water cycle). Therefore climate effects within this ESM from irrigation expansion can only occur due to increased vegetation growth as a consequence of the ample water availability (Döscher et al., 2022).

Additionally we made it more explicit in the discussion section 4.1 in order to highlight that the irrigation induced climate effects are only caused by greening due to increased water availability:

For irrigation expansion, MPI-ESM and CESM consistently show that the increase in latent heat dominates the surface temperature response, causing a local cooling. In the current EC-EARTH setup, there is no coupling of land surface moisture by water fluxes to the atmosphere caused by irrigation are not modelled, hence the only effect on the climate is due to increased growth of crops and respective changes in physical properties.

We would still argue that the EC-EARTH results of IRR-CROP (fig 7) are useful to be shown despite knowing in advance that these climate effects will be minor due to the lack of communication of the water flux between the atmosphere and vegetation dynamics model. As there are still climate effects modelled in the greening of crops in this ESM. Furthermore, it clearly illustrates the importance of modelling this water flux in order to simulate the climate effects of irrigation.